# METTL3/METTL14 maintain human nucleoli integrity by mediating SUV39H1/H2 degradation

Yongli Shan [1,2,3,4,8] ✉, Yanqi Zhang [1,3,4,8], Yanxing Wei[2,8], Cong Zhang[1,3,4], Huaisong Lin[1,4], Jiangping He [5], Junwei Wang[1,4], Wenjing Guo [1,4], Heying Li [1,4], Qianyu Chen[1,4], Tiancheng Zhou[1,4], Qi Xing[1,3,4], Yancai Liu[1,4], Jiekai Chen [1,4] & Guangjin Pan [1,3,4,6,7] ✉

Nucleoli are fundamentally essential sites for ribosome biogenesis in cells and formed by liquid-liquid phase separation (LLPS) for a multilayer condensate structure. How the nucleoli integrity is maintained remains poorly understood. Here, we reveal that METTL3/METTL14, the typical methyltransferase complex catalyzing N6-methyladnosine (m⁶A) on mRNAs maintain nucleoli integrity in human embryonic stem cells (hESCs). METTL3/METTL14 deficiency impairs nucleoli and leads to the complete loss of self-renewal in hESCs. We further show that SUV39H1/H2 protein, the methyltransferases catalyzing H3K9me3 were dramatically elevated in METTL3/METTL14 deficient cells, which causes an accumulation and infiltration of H3K9me3 across the whole nucleolus and impairs the LLPS. Mechanistically, METTL3/METTL14 complex serves as an essential adapter for CRL4 E3 ubiquitin ligase targeting SUV39H1/H2 for polyubiquitination and proteasomal degradation and therefore prevents H3K9me3 accumulation in nucleoli. Together, these findings uncover a previously unknown role of METTL3/METTL14 to maintain nucleoli integrity by facilitating SUV39H1/H2 degradation in human cells.

Nucleoli are fundamentally essential sites for ribosome biogenesis in eukaryotic cells and their malfunction leads to various severe diseases such as ribosomopathies characterized by reduced cellular metabolism and growth defect or cancers as well[1-4]. The primary roles of nucleoli in ribosome biogenesis include rRNA transcription driven by RNA polymerase I (Pol I), rRNA processing, initial ribosome assembly, etc.[5]. Nucleoli are formed around ribosome gene (rDNA) repeats arrays (nucleolar organization regions, NORs) and contain hundreds of proteins and small RNAs for rRNAs processing[5]. Nucleoli undergo dynamic cycle of assembling during normal cell cycle or under cellular stresses[6]. Although the nucleoli are the most prominent and physically separated structure in nuclei, they are not covered by a membrane[7]. Substantial evidence support that the dynamic assembling of nucleoli is via a liquid-liquid phase separation (LLPS) mechanism to form a

[1]Key Laboratory of Immune Response and Immunotherapy, Joint School of Life Sciences, Guangzhou Institutes of Biomedicine and Health, Chinese Academy of Sciences, Guangzhou Medical University, Guangzhou, China. [2]Nanfang Hospital, Southern Medical University, Guangzhou, China. [3]University of Chinese Academy of Sciences, Beijing, China. [4]Guangdong Provincial Key Laboratory of Stem Cell and Regenerative Medicine, Guangdong-Hong Kong Joint Laboratory for Stem Cell and Regenerative Medicine, Center for Cell Lineage and Cell Therapy, Guangzhou Institutes of Biomedicine and Health, Chinese Academy of Sciences, Guangzhou, China. [5]Guangzhou Laboratory, Guangzhou, China. [6]Centre for Regenerative Medicine and Health, Hong Kong Institute of Science and Innovation, Chinese Academy of Sciences, Hong Kong, Hong Kong. [7]GIBH-HKU Guangdong-Hong Kong Stem Cell and Regenerative Medicine Research Centre, GIBH-CUHK Joint Research Laboratory on Stem Cell and Regenerative Medicine, Guangzhou Institutes of Biomedicine and Health, Chinese Academy of Sciences, Guangzhou, China. [8]These authors contributed equally: Yongli Shan, Yanqi Zhang, Yanxing Wei. ✉e-mail: shan_yongli@gibh.ac.cn; pan_guangjin@gibh.ac.cn

multilayer condensate of different biomolecules[3,8,9]. Critical nucleolar proteins such as fibrillarin (FBL), Nucleophosmin (NPM1), etc. contain structures prone to phase separation that allow them to partition into distinct layers together with their associated rRNAs and proteins[10,11]. Three layers or sub-compartments (fibrillar center (FC), dense fibrillar component (DFC), and granular component (GC)) could be observed in nucleoli by electron microscopy and each of them contains proteins involved in specific stage of ribosome biogenesis[7]. rRNA transcription and processing mainly occur in FC and DFC layers while the initial ribosome assembly happens in GC[3]. The liquid property and LLPS play essential roles to maintain nucleoli integrity while disruption of LLPS impairs nucleoli structure and leads to severe cell defects and diseases[9,12,13]. However, how the liquidity and LLPS are maintained in nucleolus, particularly in such a membrane-less sub-nuclear organelle remains less understood.

In human cells, the mature nucleoli are surrounded with peri-nucleolar heterochromatin (PNH) derived from DNA sequences located distal or proximal to rDNA NORs[14,15]. Heterochromatin is typically associated with the repressive histone H3K9 trimethylation (H3K9me3) that is catalyzed by SUV39H1/2 methyltransferases[16]. Recently, the heterochromatin binding protein, HP1 was reported to regulate nucleolar structure in mouse embryonic stem cells (ESCs), indicating that the disorganization of PNH impacts nucleolar structure[10]. However, it remains unclear how the PNH is placed and well-controlled to maintain the normal nucleolar structure.

METTL3/METTL14 form a conventional methyltransferase complex (MTC) that catalyzes N6-methyladnosine (m⁶A) on mRNAs and regulates various biological processes[17–21]. m⁶A modification on RNAs marks wide-ranging transcripts in mammalian and involves in diverse RNA metabolism process, including RNA stability, splicing, transport, and translation, etc.[22–24]. The non-conventional roles of MTC have been reported to regulate heterochromatin and silence retroviral elements via H3K9me3 in mouse embryonic stem cells (mESCs)[25–27].

Here in this study, we reveal a previously unknown mechanism that METTL3/METTL14 maintain nucleoli integrity in human embryonic stem cells (hESCs). We demonstrate that METTL3/METTL14 serve as an essential adapter for CRL4 E3 ubiquitin ligase that targets H3K9me3 methyltransferases SUV39H1/H2 for polyubiquitination and proteasomal degradation, which therefore prevents H3K9me3 accumulation in nucleoli and maintain their normal LLPS and structure.

## Results

### Loss of METTL3/METTL14 impairs nucleoli integrity in human ESCs

To investigate the role of METTL3/METTL14, we knocked out METTL3 or METTL14 in human ESCs, as validated by genomic PCR and western blot (Fig. 1a and Supplementary Fig. 1a, b). METTL3⁻/⁻ and METTL14⁻/⁻ hESCs showed the same phenotype with greatly reduced or halted self-renewal and enlarged cell body containing multiple nuclei (Fig. 1a-b and Supplementary Fig. 1c). However, based on transcriptome analysis, the expression of typical pluripotency genes was well maintained while the lineage genes were not induced, indicating that METTL3 or METTL14 deletion uncouples self-renewal from pluripotency in human ESCs (Fig. 1c and Supplementary Fig. 1d-f). Cell cycle analysis showed reduced S phase and increased G2/M phase cells in mutant hESCs (Fig. 1d), supporting the observed growth defect in these cells. There was no significant difference on cell apoptosis between wild type (WT) and mutant hESCs (Supplementary Fig. 1g). The down-regulated genes in METTL3⁻/⁻ and METTL14⁻/⁻ hESCs were enriched in functions related to cellular metabolism, translation, rRNA processing as well as cell cycle progression etc. (Fig. 1c), indicating a biosynthesis dysfunction in METTL3⁻/⁻ and METTL14⁻/⁻ hESCs.

Interestingly, we observed a striking nucleolar change in METTL3⁻/⁻ and METTL14⁻/⁻ hESCs (Fig. 1e). WT hESCs usually had 1–2 nucleoli per cell with relatively large size that are around 6–7 µm

diameter and 30 µm² (Fig. 1e, f). In contrast, METTL3⁻/⁻ or METTL14⁻/⁻ hESCs showed substantially increased number of nucleoli, but with much reduced size (Fig. 1e–f). To further examine the nucleoli functions, we analyzed rRNA transcripts by specific primers targeting various rRNA regions (Fig. 1g). rRNA transcription showed substantial defect in METTL3⁻/⁻ and METTL14⁻/⁻ hESCs (Fig. 1g, Supplementary Fig. 1h). Since rRNA represents the majority of total RNA in the cells, accordingly, the nascent RNA was much reduced in mutant hESCs (Supplementary Fig. 1i). However, the expression of nucleolar genes such as NPM1, FBL, and UBF was not changed in mutant cells (Supplementary Fig. 1j). These data indicate a functional defect in nucleoli of METTL3⁻/⁻ and METTL14⁻/⁻ hESCs.

We then examined the mature ribosome assembly in METTL3/METTL14 deficient cells. Since METTL3⁻/⁻ and METTL14⁻/⁻ hESCs stopped self-renewal and were hard to produce enough cells for ribosome profiling, we introduced a DOX inducible METTL3 expression in METTL3⁻/⁻ hESCs (METTL3-OE/KO) to rescue the cell defect (Supplementary Fig. 2a–i). The normal phenotype in METTL3⁻/⁻ hESCs was well maintained by DOX treatment (Supplementary Fig. 2b–h). Upon DOX withdrawal, METTL3 gradually disappeared and the cells stopped self-renewal and showed nucleoli defect (Supplementary Fig. 2b–i, Supplementary Fig. 3a–c). Based on this cell model, we showed that the mature ribosome assembly was greatly reduced in the absence of METTL3 (Fig. 1h, Supplementary Fig. 3d), indicating that MTC is essential to maintain nucleoli function for ribosome assembly. Disruption of nucleoli structure has been known to induce a so-called "nucleolar stress" that triggers cell surveillance system, such p53 activation[1,28]. Indeed, the protein level of P53 and the mRNA level of P53 pathway genes greatly increased in METTL3⁻/⁻ and METTL14⁻/⁻ hESCs (Fig. 1i, Supplementary Fig. 3e, f), indicating that the surveillance system was activated to block self-renewal in these cells. Together, our data reveal an essential role of METTL3/METTL14 to maintain nucleoli integrity and self-renewal in human ESCs.

### METTL3/METTL14 maintain phase separation in nucleoli

To extensively investigate the role of METTL3/METTL14 in nucleoli regulation, we further examined the assembly of nucleoli during cell cycle progress. We introduced a lenti-viral based expression of NPM1-GFP and H2B-mCherry fusion protein into the cells and performed live cell imaging to monitor nucleoli assembly (Fig. 2a). WT hESCs showed the typical coalescence of small nucleoli into 1 or 2 big and mature nucleoli during cell cycle progress (Fig. 2a). In contrast, METTL3-deficient hESCs showed substantial defect in normal nucleoli coalescence (Fig. 2a). Since liquid-liquid phase separation (LLPS) serves a critical mechanism to form multilayer condensate for nucleolar structure, we examined structure of three major layers or compartment in nucleoli. In WT hESCs, the FBL labeled DFC, NPM1 labeled GC as well as UBF labeled FC were clearly separated (Fig. 2b, c, upper panel). In contrast, in METTL3⁻/⁻ or METTL14⁻/⁻ hESCs, these nucleolar proteins were largely inter-mixed to be an irregular structure and no clear phase separation in nucleoli could be observed (Fig. 2b, c, lower panel). These data demonstrate that METTL3/METTL14 complex is essential to maintain phase separation in nucleoli to form the multi-layer condensate.

The liquidity property of nucleoli is critical for the nucleolar biomolecules to mobile and form specific layers within nucleolus. Based on the Fluorescence Recovery after Photobleaching (FRAP) assay, GFP labeled NPM1 protein showed much reduced recovery and mobility in METTL3-deficient hESCs (METTL3-OE/KO/DOX-) compared with WT and rescued cells (Fig. 2d), indicating that the nucleolar liquidity was largely compromised in mutant cells. Lastly, treatment of 1,6-hexanediol (1,6-HD), an aliphatic alcohol known to disrupt hydrophobic interactions and thus LLPS[10,29], generated nucleolar defect in WT hESCs phenocopied METTL3⁻/⁻ and METTL14⁻/⁻ hESCs (Supplementary Fig. 4a–d). All these data demonstrate that METTL3/METTL14 complex

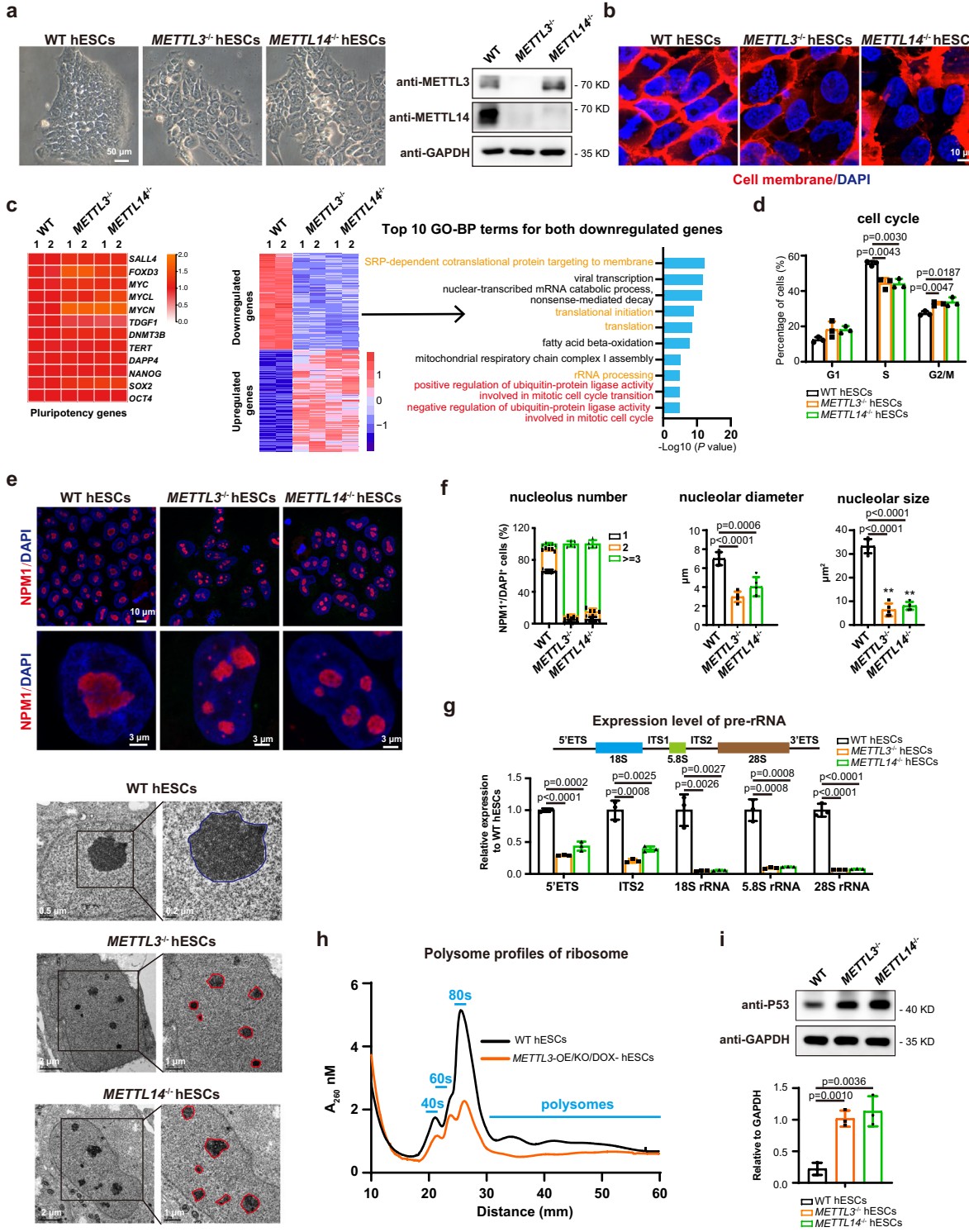

maintains liquidity and phase separation in human nucleoli for forming the normal multilayer condensate.

## METTL3/METTL14 prevent the nucleolar heterochromatinization

The mature nucleoli were known to be surrounded by peri-nucleolar heterochromatin (PNH) labeled by H3K9me3 that was catalyzed by SUV39H1/2 enzymes[10,16]. METTL3 was reported to regulate heterochromatin and silence retroviral elements (REs) via H3K9me3[26]. *Mettl3* knock-out in mouse ESCs showed a reduced H3K9me3 and activation

of REs[26]. Surprisingly, in this study, H3K9me3 greatly increased in *METTL3*[−/−] and *METTL14*[−/−] hESCs compared with WT cells (Fig. 3a). In WT hESCs, the dense H3K9me3 signal mainly localized around the nucleolar while the nucleolar itself contained little or no H3K9me3 signal and showed much loose chromatin (Fig. 3b). However, in *METTL3*[−/−] or *METTL14*[−/−] hESCs, H3K9me3 signal greatly increased and infiltrated into the nucleoli compared with other regions in nucleus (Fig. 3b). Compared with WT nucleoli that contained little H3K9me3, nucleoli in *METTL3*[−/−] and *METTL14*[−/−] hESCs were filled with H3K9me3 (Fig. 3b). The H3K9me3 enzymes, SUV39H1 or SUV39H2 also

**Fig. 1 | METTL3/METTL14 knock-out impairs nucleoli integrity and function in human ESCs. a** Left panel, Representative images for morphology of wild type (WT), *METTL3-* or *METTL14-* deleted H1 hESCs from three independent images (n = 3). Scale bar, 50 μm. Right panel, western blot of METTL3 or METTL14 in indicated cells. **b** Membrane staining of the indicated cells. The cells were stained by DiD for membrane and DAPI for nuclei. Scale bar, 10 μm. **c** Left panel, Heatmap on the expressions of selected pluripotent genes in the indicated cell lines. Middle panel, Heatmap of down-regulated genes or up-regulated genes in *METTL3-* or *METTL14*-deficient hESCs compared with WT H1 hESCs. Right panel, top 10 enriched GO-BP terms in downregulated genes. **d** Cell cycle analysis of wild type (WT), *METTL3*-deficient, and *METTL14*-deficient hESCs. **e** Upper panel, immunostaining on the nucleolus marker NPM1 in the indicated cell lines. Scale bar, 10 μm, and 3 μm. Lower panel, transmission electron micrographs of nuclear and nucleolar morphology of WT, *METTL3*-deficient, or *METTL14*-deficient hESCs. Nucleoli regions were enlarged as indicated. Scale bars were indicated. **f** The quantification of nucleolus number, nucleolar diameter, and size in the indicated cell lines. The nucleoli were labeled by NPM1 and calculated manually. **g** qRT-PCR analysis on the expression of pre-rRNA in the indicated cell lines. Wild type H1 hESCs serve as control. **h** Polysome profiling in wild type (WT) hESCs and DOX dependent exogenous *METTL3* expression in *METTL3⁻/⁻* hESCs (*METTL3*-OE/KO) upon withdrawal of DOX treatment at day 18. The profiling was analyzed using by sucrose density-gradient ultracentrifugation. **i** Western blot analysis and quantification on P53 the indicated cell lines. The data in (**d**, **g**, and **i**) represent mean ± SD from three independent experiments (n = 3), and in (**f**) represent mean ± SD from five independent experiments (n = 5). Statistical analysis, unpaired two-tailed Student's t-test. **P < 0.01.

showed substantial increase in *METTL3⁻/⁻* and *METTL14⁻/⁻* hESCs (Fig. 3c). Consistent to H3K9me3, SUV39H1/2 mainly localized around nucleoli in WT cells but penetrated into and filled the nucleoli in *METTL3⁻/⁻* and *METTL14⁻/⁻* hESCs (Fig. 3d-e). These data indicate that METTL3/METTL14 prevent accumulation of SUV39H1/2 in nucleoli to maintain their normal structure. Indeed, overexpression of SUV39H1/2 and not SETDB1 generated similar nucleoli defects in hESCs (Supplementary Fig. 5a-e). Taken all the data together, we reveal that METTL3/METTL14 prevent the over-heterochromatinization in human nucleoli to maintain their integrity. In addition, we also examined the chromatin localization of H3K9me3 in WT and mutant hESCs (Supplementary Fig. 5f-i). H3K9me3 showed an obviously increased enrichment in *METTL3⁻/⁻* hESCs compared with WT cells, indicating METTL3 also prevents global heterochromatinization in hESCs.

## Core motifs of METTL3/14 are essential to maintain nucleoli integrity

Conventionally, METTL3/METTL14 form a m⁶A methyltransferase complex to modify m⁶A on mRNAs[30,31]. Our data shown above indicate that METTL3 and METTL14 are not functionally redundant in maintaining nucleoli integrity. The core motifs essential for the biological function of METTL3 or METTL14 had been identified in previous reports, such as DPPW motif for METTL3 or EPPL motif for METTL14[20,27,32]. We then performed a rescue experiment using either WT or mutant forms (MUT) of these motifs in METTL3 or METTL14 based on the afore-described DOX inducible exogenous METTL3 or METTL14 expression in *METTL3⁻/⁻* or *METTL14⁻/⁻* hESCs (*METTL3*-OE/KO and *METTL14*-OE/KO hESCs, respectively) (Supplementary Fig. 2a−h and Fig. 4a, Supplementary Fig. 6a). Upon DOX withdrawal, these cells showed complete loss of self-renewal while were fully rescued by the WT METTL3 or METTL14 (Fig. 4a, b, Supplementary Fig. 6b). However, the mutant forms of either METTL3 or METTL14 failed to rescue the growth defect in *METTL3*-OE/KO/DOX- and *METTL14*-OE/KO/DOX- hESCs (Fig. 4a, b, Supplementary Fig. 6b). WT METTL3 or METTL14 localized in nucleoli and fully rescued the nucleolar structure and functional defects in terms of rRNA expression and mature ribosome assembly in the corresponding knock-out cells (Fig. 4b–d, Supplementary Fig. 6b, c). However, the mutant METTL3 or METTL14 failed to rescue the nucleoli structure and function, such as rRNA expression and mature ribosome assembly (Fig. 4b–d, Supplementary Fig. 6b, c). Accordingly, LLPS to form different layers in nucleoli was completely restored by WT METTL3 or METTL14, but not the mutant forms (Supplementary Fig. 6d, e). Lastly, WT METTL3 or METTL14 successfully prevented infiltration of H3K9me3 in nucleoli, but the mutant forms completely lost this function (Supplementary Fig. 6f). On the other hand, nucleolar proteins usually contain a nucleolar localization signal[33,34]. Indeed, we did identify this signal motif in METTL3 (Fig. 4e). Mutation of this signal motif in METTL3 (*METTL3^△NoLS^*) clearly impaired its nucleolar localization and the function to rescue the defect in *METTL3*-deficient cells (Fig. 4e–g, Supplementary Fig. 6g). Together, these data demonstrate that the core motifs of METTL3/METTL14 are essential for their function to maintain nucleoli LLPS and integrity.

## METTL3/METTL14 mediate SUV39H1/H2 proteasomal degradation

The substantial elevation of SUV39H1/H2 protein, while not their mRNAs in *METTL3* or *METTL4* knockout cells (Fig. 3c and Fig. 5a) prompted us to further investigate the protein stability in these cells. Upon inhibition of protein synthesis in WT hESCs by cycloheximide (CHX), SUV39H1 or SUV39H2 protein underwent rapid degradation within 6 h (Fig. 5b, Supplementary Fig. 7a). Accordingly, H3K9me3 disappeared even more rapidly within 3 h (Fig. 5b, Supplementary Fig. 7a). In big contrast, SUV39H1 or SUV39H2 protein failed to degrade and the H3K9me3 persisted in *METTL3* or *METTL4* knockout cells after protein synthesis inhibition (Fig. 5b, Supplementary Fig. 7a). Proteasome serves a major mechanism to degrade proteins that are ubiquitinated in the cell. Inhibition of proteasome activity largely blocked SUV39H1 or SUV39H2 protein degradation in hESCs (Fig. 5c, Supplementary Fig. 7b). All these data indicate that SUV39H1 or SUV39H2 protein were subject to ubiquitination-dependent proteasomal degradation and METTL3/METTL4 complex is essential in this process.

We then sought to directly examine whether SUV39H1 or SUV39H2 is indeed ubiquitinated and whether METTL3/METTL14 complex is involved in their ubiquitylation. We firstly generated expression vectors of SUV39H1 or SUV39H2, METTL3, mutant METTL3, and ubiquitin with different tags and performed co-transfection and co-immunoprecipitation (co-IP) of these factors (Fig. 5d). As shown in Fig. 5d, the FLAG-tagged SUV39H1 or SUV39H2 was efficiently pulled down and detected by anti-FLAG antibody (Fig. 5d). The MYC-tagged ubiquitin on SUV39H1 or SUV39H2 were clearly detected by anti-MYC antibody (Fig. 5d, lane 2−4). HA-tagged METTL3 was co-precipitated with flag-tagged SUV39H1 or SUV39H2 (Fig. 5d, lane 2). Notably, SUV39H1 or SUV39H2 co-transfected with WT METTL3 showed more ubiquitylation compared with SUV39H1 or SUV39H2 transfected alone or with the mutant METTL3 (Fig. 5d, compare lane 2 with lane 3 and 4). These data demonstrate that METTL3 associates with SUV39H1/H2 and promote their ubiquitylation.

## METTL3/14 serve as an adapter for CRL4 E3 ubiquitin ligase

In ubiquitin pathway, the E3 ubiquitin ligase is responsible for recognizing a specific target and catalyzing ubiquitylation[35]. The cullin family proteins are evolutionarily conserved and can bind to a small RING protein to assemble a large family of cullin-RING E3 ligases (CRLs) that target around 20% of total proteins for proteasomal degradation[36,37]. Unlike other RING E3 ubiquitin ligases, CRLs do not directly bind targets but rather rely on adapters to specifically recognize their targets[35,38]. Among CRLs, the cullin 4-RING ligase (CRL4) was reported to regulate SUV39H1 in zygote, based on the binding of CRL4 linker protein, DDB1, and the adapter, DDB1/CUL4 associated factors (DCAFs)[36]. Since that METTL3/14 deficiency impaired SUV39H1 and SUV39H2 proteasomal degradation (Fig. 5b, c) and METTL3 promotes

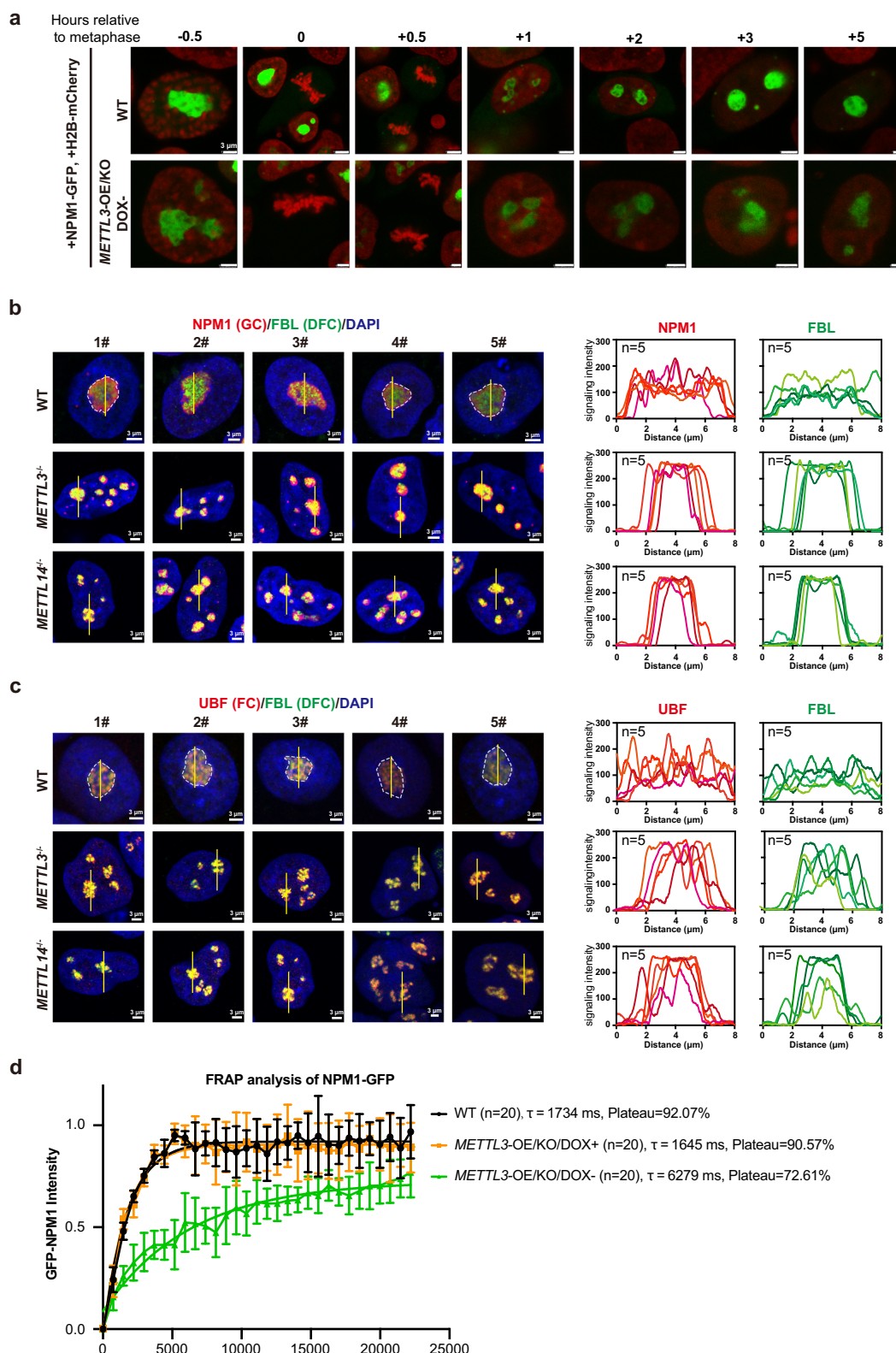

their ubiquitylation (Fig. 5d), we reasoned that METTL3/14 complex might serve as a previously unknown adapter for CRL4 targeting SUV39H1 and SUV39H2 for their proteasomal degradation. We then performed co-IP assay in WT or *METTL3*⁻/⁻ hESCs as well as *METTL3*⁻/⁻ hESCs rescued by WT or mutant METTL3 (Fig. 5e). Based on co-IP assay by anti-METTL3 antibody, SUV39H1/H2 and CRL4 linker DDB1 could be clearly detected to interact with METTL3 in WT hESCs (Fig. 5e upper

panel, lane 1, Supplementary Fig. 7c) or *METTL3*⁻/⁻ hESCs rescued by WT METTL3 (Fig. 5e upper panel, lane 3, Supplementary Fig. 7c), but their interaction with mutant METTL3 was much weaker (Fig. 5e upper panel, lane 4, Supplementary Fig. 7c). On the other hand, based on co-IP assay by anti-DDB1 antibody, SUV39H1/H2 interacted with DDB1 and METTL3 in WT hESCs (Fig. 5e middle panel, lane 1, Supplementary Fig. 7d) or *METTL3*⁻/⁻ hESCs rescued by WT METTL3 (Fig. 5e middle

**Fig. 2 | METTL3/METTL14 knock-out reduces nucleolar liquidity and disrupts phase separation in nucleoli. a** Live cell imaging of WT and *METTL3*-deficient hESCs. Lentiviral-based NPM1-GFP and H2B-mCherry were transfused to WT or *METTL3*-OE/KO/DOX- (deficient) hESCs and imaged by SP8-STED. Time is presented when the metaphase of mitosis is observed. Scale bar, 3 μm. **b** Left panel: five representative immunostaining (n = 5) on FBL (DFC marker) and NPM1 (GC marker) in WT, *METTL3*-deficient or *METTL14*-deficient hESCs. Scale bar, 3 μm. Right panel, line scans depict signaling strength of NPM1 (red) and FBL (green) across nucleolus. **c** Left panel: five representative immunostaining (n = 5) on FBL (DFC marker) and UBF (FC marker) in WT, *METTL3*-deficient, or *METTL14*-deficient hESCs. Scale bar, 3 μm. Right panel, line scans depict signaling strength of UBF (red) and FBL (green) across nucleolus. **d** The intensity dynamics of the bleached regions within NPM1-GFP condensates (n = 20) was fit to an exponential function during FRAP of NPM1-GFP within a region of one nucleolus in wild type (WT), *METTL3*-OE/KO/DOX+ (rescued), and *METTL3*-OE/KO/DOX- (KO) hESCs. The data represent mean ± SD from 20 independent experiments (n = 20).

panel, lane 3, Supplementary Fig. 7d), but the interaction between SUV39H1/H2 and DDB1 was completely impaired in *METTL3⁻/⁻* hESCs (Fig. 5e middle panel, lane 2, Supplementary Fig. 7d) or reduced in *METTL3⁻/⁻* hESCs rescued by mutant METTL3 (Fig. 5e middle panel, lane 4, Supplementary Fig. 7d). Similarly, the interaction of METTL14 and DDB1 could also be detected by co-IP assay (Fig. 5e middle panel). These interactions could also be detected by FRET assay (Supplementary Fig. 7e–g). Lastly, since METTL3/14 is an RNA binding complex, we showed that the interactions of METTL3 with DDB1 and SUV39H1/2 were impaired by RNase A treatment (Supplementary Fig. 8a). The interaction of DDB1 and SUV39H1/H2 was also impaired upon the inhibition of nascent RNA synthesis with Actinomycin D (Fig. 5f and Supplementary Fig. 8b–e). Accordingly, SUV39H1/H2 protein as well as H3K9me3 were greatly accumulated after nascent RNA inhibition (Fig. 5f lower panel and Supplementary Fig. 8b–e). Taken all these data together, we reveal that METTL3/14 complex serves as an essential adapter for CRL4 targeting SUV39H1/H2 for proteasomal degradation and prevents H3K9me3 accumulation and infiltration into nucleoli.

## Discussion

Here in this study, we uncover an essential role of METTL3/METTL14, the conventional m⁶A methyltransferase complex to maintain nucleolar integrity and human ESC self-renewal. METTL3/ METTL14 seclude over-heterochromatinization in nucleoli and maintain their normal liquidity to allow the normal nucleolar assembly via LLPS mechanism. We reveal a previously unknown role of METTL3/ METTL14 complex as an essential adapter for CRL4 E3 ubiquitin ligase targeting SUV39H1/H2 for proteasomal degradation, therefore preventing H3K9me3 accumulation across the nucleoli (Fig. 5g).

As a membrane-less and fundamentally essential cell organelle, the mature nucleoli were assembled via a LLPS to form multilayer condensate of different biomolecules prone to phase separation[3,33,39]. The integrity of multilayer structure is critical to nucleolar function because each layer is involved in different and specific stage of ribosome biogenesis[40]. The mature nucleoli are surrounded with peri-nucleolar heterochromatin (PNH) derived from DNA sequences located distal or proximal to rDNA NORs[15,41]. The exact role of PNH in nucleolar function and integrity remains less clear. As the most active sites for transcription, nucleoli exhibit a much loose chromatin structure (Fig. 1 and other Figures for nuclear DAPI staining images). In absence of METTL3/METTL14, nucleolar chromatin is catalyzed with H3K9me3 and become heterochromatin that reduces liquidity and impairs LLPS. Recently, in another report, deletion of HP1, the specific heterochromatin protein also impairs nucleolar structure in mouse ESCs[10]. These data indicate that PNH regulation is critical to maintain nucleolar integrity. These data indicate that METTL3/METTL14 prevent the nucleolar heterochromatinization depending on the nascent rRNAs (Fig. 5g).

Conventionally, METTL3 and METTL14 form a typical RNA methyltransferase complex to catalyze m⁶A on mRNAs[30,42]. Here, our data uncover a non-conventional role of METTL3/14 in mediating SUV39H1/2 degradation and thus maintaining nucleoli integrity dependent on their critical motifs, such as the previously unrecognized nucleolar localization signal motif (Fig. 4). However, the precise function of m⁶A modification on rRNA in this process is worth to investigate in future projects. On the other hand, we performed KO of *YTHDC1*, the typical reader for m⁶A on mRNAs in hESCs, but did not observe a significant phenotype change, indicating a differential mechanism or factors to read m⁶A modifications on mRNA and rRNA. In addition, the functions of other writer or eraser proteins in this process such as WTAP/KIAA1429, ALKBH5/FTO, ZCCHC4, and METTL5[21,32,43,44] also would be worth to pursue in future projects.

Our finding that METTL3/METTL14 complex serves as an essential adapter for CRL4 to target SUV39H1/H2 for ubiquitylation of proteasomal degradation is intriguing. To prevent over-spreading of H3K9me3 across the genome, SUV39H1/H2 are subject to rapid degradation (Fig. 5b). METTL3/METTL14 deficiency leads substantial elevation of SUV39H1/H2 protein and H3K9me3 that infiltrates across the whole nucleoli (Fig. 3). CRLs is a group of special E3 ubiquitin ligases that rely on adapters or substrate-recruiting receptors for the target specificity[35]. Assembled by CUL4 and the linker DDB1, CRL4 form more than 90 E3 complexes in mammals and many of them are involved in chromatin regulation[38]. The well-studied adapter for CRL4 are DDB1-CUL4-associated factors (DCAFs) that usually contain WD40 repeats[37]. DCAF13 was reported to recruit SUV39H1/H2 for CRL4 in mouse zygotes and promote zygotic gene expression[36]. Our data demonstrate that METTL3/14 complex serves as an essential adapter in human ESCs for CRL4 targeting SUV39H1/H2. It's quite unexpected since that METTL3 is not a member of typical DCAFs that contain WD40 repeats, but rather depending on conserved core motifs to link to CRL4. Indeed, either mutation of their core motifs or abolishing nascent RNA impaired METTLL3/14-mediated SUV39H1/H2 degradation (Figs. 4 and 5, Supplementary Fig. 8). Our study provides another mechanism how RNAs and RNA binding protein (RBPs) impact chromatin structure, particular for the nucleolar chromatin that is highly active for rRNA transcription. Notably, knockdown of *METTL3* in HEK293T and HeLa cells did not induce a significant change of SUV39H1/2 proteins (Supplementary Fig. 8g, h), indicating that the role of METTL3/METTL14 in mediating SUV39H1/2 degradation might be not universal. The exact role of METTL3/14-SUV39H1/H2-nucleoli axis would definitely be worth to investigate in more model systems, such as tumorigenesis.

## Methods

### Cell culture

The use of human H1 ESCs (WiCell, hPSCReg ID: WAe001-A) in this study was approved by the Life Science and Medical Ethics Committee at the Guangzhou Institutes of Biomedicine and Health, Chinese Academy of Sciences. The culture plates coated with Matrigel (Corning) were used to culture the human embryonic stem cell lines H1 and its derived cell lines. These cell lines are cultured in mTeSR1 medium (STEMCELL Technologies) at 37 °C with 5% CO₂ under mycoplasma-free and sterile conditions. The fresh mTeSR1 medium was changed every day and the hESCs were passaged every 3 days using 0.5 mM EDTA.

### Gene editing in human ESCs

For gene knockout[45], we respectively designed specific guide RNAs (gRNAs) targeting *METTL3* and *METTL14* on the website https://crispr.cos.uni-heidelberg.de/ and inserted these gRNAs into pX330 (Addgene) vector. Meanwhile, homologous arm donor vectors

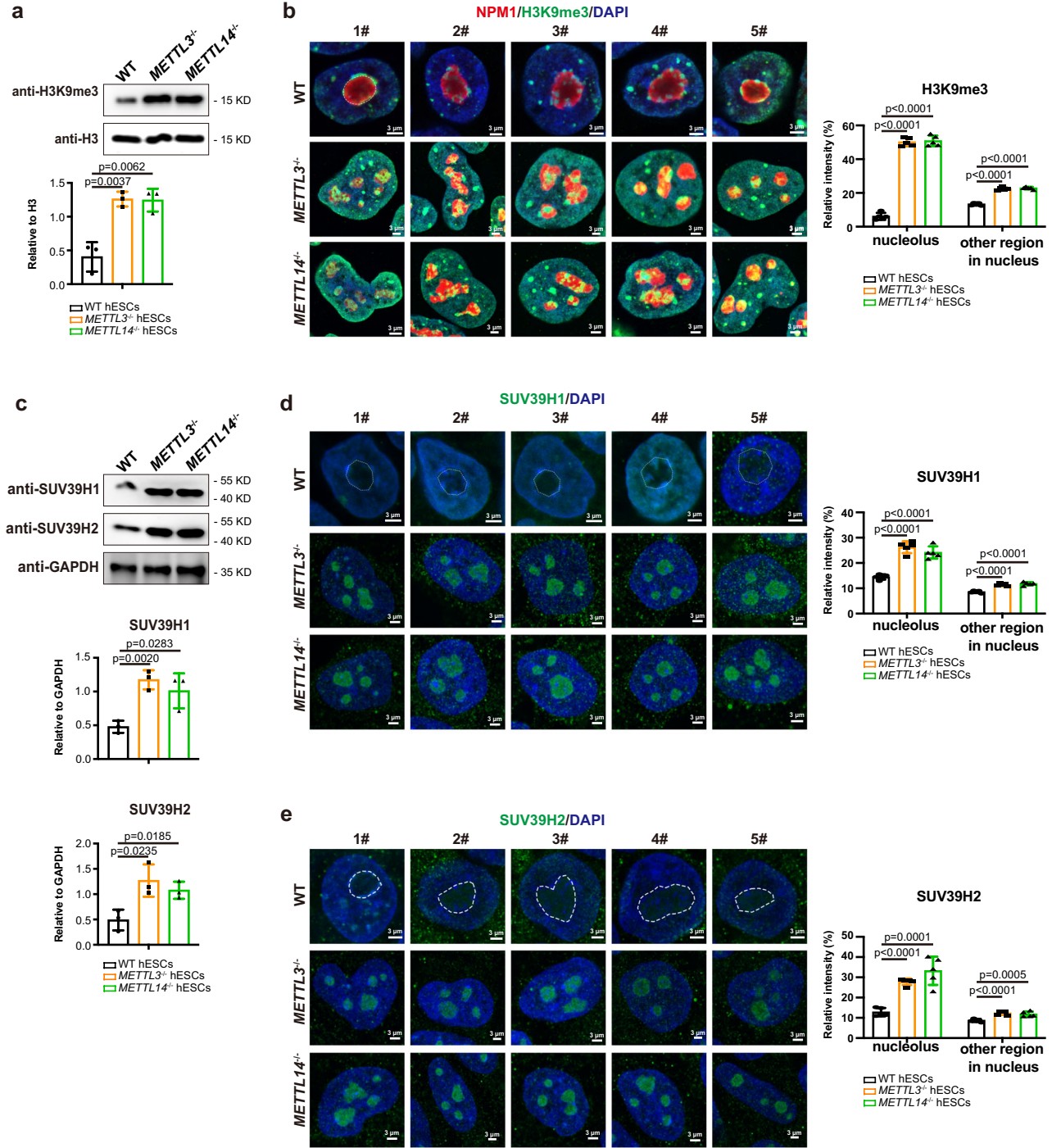

**Fig. 3 | Infiltration of heterochromatin in nucleoli in METTL3/METTL14-deficient hESCs. a** Western blot analysis and quantification for H3K9me3 in WT, *METTL3*-deficient, and *METTL14*-deficient hESCs. **b** Left panel: five representative immunostaining on H3K9me3 and nucleolar marker NPM1 in the indicated cells. Scale bar, 3 μm. Right panel: the quantification of fluorescence intensity of H3K9me3 in nucleolus and other region in nucleus. **c** Western blot analysis and quantification for SUV39H1 and SUV39H2 in the indicated cell lines. **d** Left panel: immunostaining on SUV39H1 in the indicated cell lines. Scale bar, 3 μm. Right panel: the quantification of fluorescence intensity of SUV39H1 in nucleolus and other region in nucleus. **e** Left panel: immunostaining on SUV39H2 and NPM1 in the indicated cell lines. Scale bar, 3 μm. Right panel: the quantification of fluorescence intensity of SUV39H2 in nucleolus and other region in nucleus. The data in (**a**, **c**) represent mean ± SD from three independent experiments (n = 3), and in (**b**, **d**, and **e**) represent mean ± SD from five independent experiments (n = 5). Statistical analysis, unpaired two-tailed Student's t-test.

targeting *METLL3* and *METTL14* contained left and right homologous arms from these genes and a LoxP-flanked PGK-puromycin cassette. We collected 1 million hESCs and transferred 4 μg donor DNA and 4 μg pX330 plasmid containing the corresponding gRNA into hESCs by electroporation. These electroporated hESCs were cultured in mTeSR1 plus 0.5 μM Thiazovivin (selleck) for 1 day. Then, the positive clones for targeting *METLL3* and *METTL14* were selected using 2 μg/mL Puromycin (Gibco). For gene knockout, morphological changes were

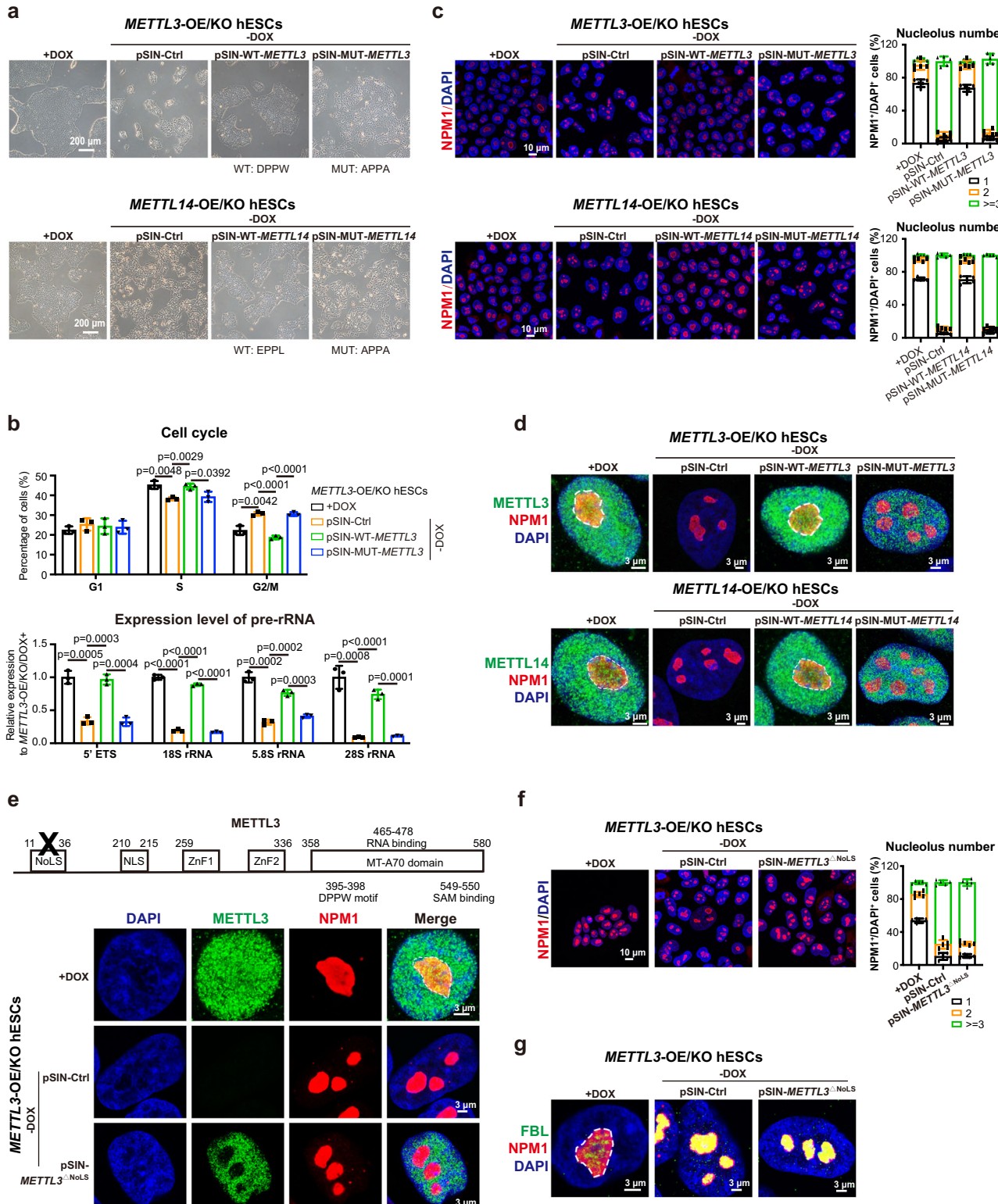

**Fig. 4 | METTL3/METTL14 maintain nucleoli integrity depending on their core motifs. a** Upper panel, Morphology of *METTL3* deficient cells (*METTL3*-OE/KO/DOX-) rescued by WT or mutant *METTL3* as indicated. Below panel, Morphology of *METTL14* deficient cells (*METTL14*-OE/KO/DOX-) rescued by WT or mutant *METTL14* as indicated. Scale bar, 200 μm. **b** Cell cycle analysis and qRT-PCR analysis on the expression level of pre-rRNA in the indicated cells in (**a**). The data represent mean ± SD from three independent experiments (n = 3). Statistical analysis, unpaired two-tailed Student's t-test. **c** Immunostaining analysis on NPM1 and the quantification of nucleolus number in the indicated cell lines in (**a**). Scale bar,

10 μm. The data represent mean ± SD from five independent experiments (n = 5). **d** Immunostaining analysis on METTL3 or METTL14 and NPM1 in indicated cells in (**a**). Scale bar, 3 μm. **e** Upper panel, the important domains in METTL3 protein. Below panel, immunostaining analysis on METTL3 and NPM1 in the indicated cell lines. Scale bar, 3 μm. **f** Immunostaining analysis on NPM1 and the quantification of nucleolus number in the indicated cell lines in (**e**). Scale bar, 10 μm. The data represent mean ± SD from five independent experiments (n = 5). **g** Immunostaining on FBL and NPM1 in the indicated cell lines in (**e**). Scale bar, 3 μm.

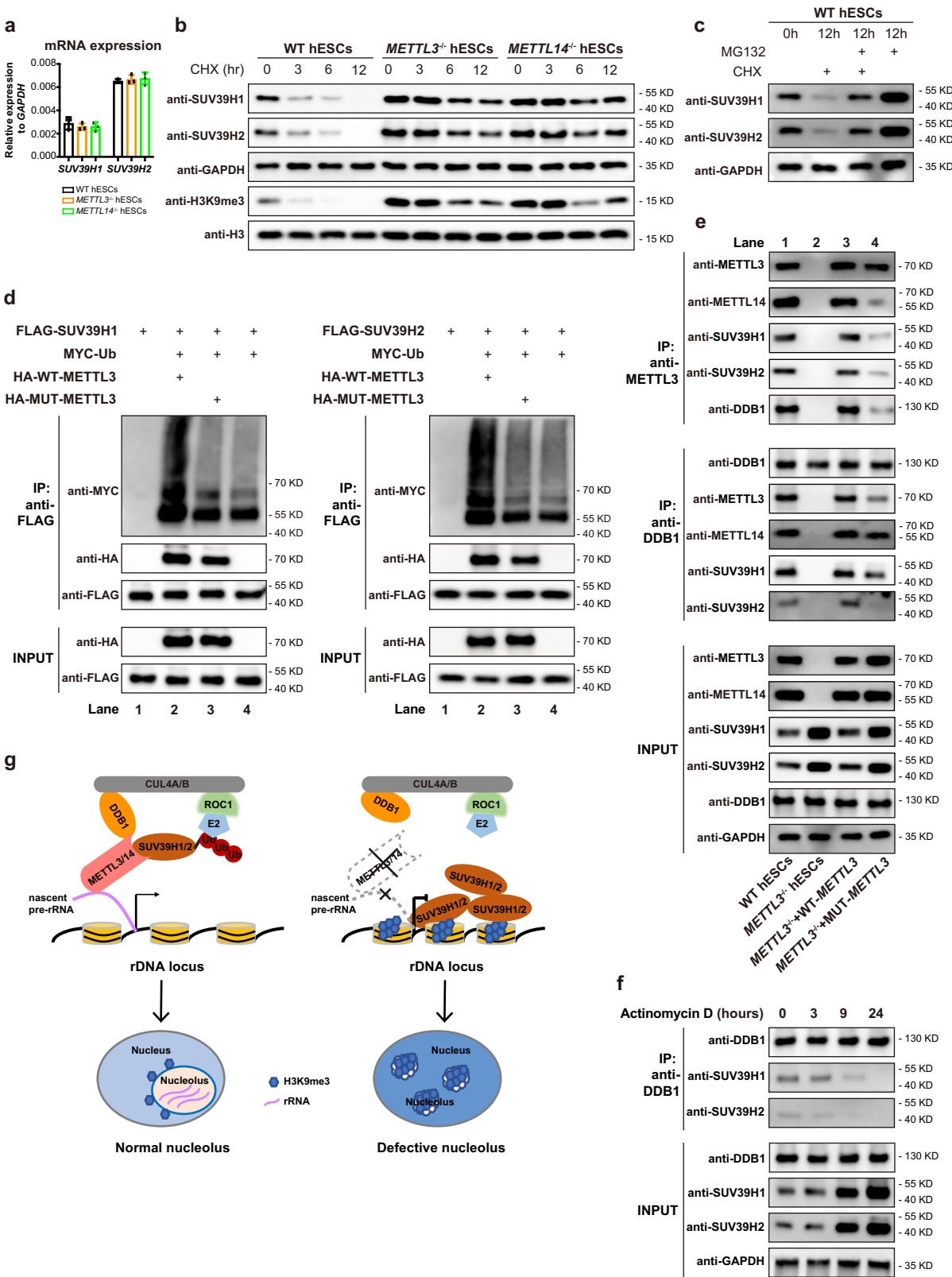

observed after continuous culture with mTeSR1 containing 2 µg/mL Puromycin for 5–6 days. Two positive clones of *METTL3-* or *METTL14-* knockout hESCs were randomly picked for genotype validation using PCR. For PCR, genomic DNAs of the clones were extracted and used for template. The F1/R1 and F2/R2 primer sets for each gene were used to amplify an about 2.5 kb product of the random integration and about 2.7 kb product of targeted integration. In addition, western blotting

was used to validate deletion of METTL3 and METTL14 in *METTL3-* or *METTL14-*knockout hESCs.

For Inducible system for gene knockout of *METLL3* and *METTL14*[46], we respectively introduced an inducible over-expression system for *METTL3* and *METTL14* in hESCs, named *METTL3*-OE and *METTL14*-OE. Then, we knocked out endogenous *METTL3* in *METTL3*-OE hESCs according to the above method. Then, the positive clones for targeting

**Fig. 5 | METTL3/METTL14 mediate SUV39H1/H2 proteasomal degradation as an essential adapter for CRL4 E3 ubiquitin ligase. a** qRT-PCR analysis on the expression of *SUV39H1* and *SUV39H2* at mRNA level in WT, *METTL3*-deficient, and *METTL14*-deficient hESCs. The data represent mean ± SD from three independent experiments. **b** Western blot results for H3K9me3, SUV39H1, and SUV39H2 in WT, *METTL3*-deficient, and *METTL14*-deficient hESCs upon cycloheximide (CHX) treatment at 3, 6, 12 h. **c** Western blot analysis for SUV39H1 and SUV39H2 in WT hESCs upon cycloheximide (CHX) and MG132 treatment at 12 h. **d** Co-IP assay by anti-flag antibody and western blot analysis for flag-tagged SUV39H1 (left panel) or SUV39H2 (right panel), MYC-tagged ubiquitin (Ub), HA-tagged WT-METTL3, HA-

tagged mutant METTL3 by corresponding anti-tag antibodies in 293 T cells. **e** METTL3/DDB1 Co-IP assay and western blot analysis for METTL3, METTL14, SUV39H1, SUV39H2, and DDB1 in WT or *METTL3⁻/⁻* hESCs as well as *METTL3⁻/⁻* hESCs rescued by WT or mutant METTL3. **f** DDB1 Co-IP assay and western blot analysis for SUV39H1 and SUV39H2 in WT hESCs upon Actinomycin D treatment at 3, 9, 24 h. **g** Schematic Model of METTL3/14 regulate nucleolar integrity in human ESCs. METTL3 serves as an essential adapter for CRL4 E3 ubiquitin ligase targeting SUV39H1/H2 for their proteasomal degradation, therefore preventing H3K9me3 accumulation across the nucleoli. The representative western blot results in (**b**–**f**) are presented from three independent experiments (n = 3).

*METTL3* and *METTL14* were selected using 2 μg/mL Puromycin for 2–3 days. Then, these clones were cultured in mTeSR1 containing 2 μg/mL doxycycline (DOX, Beyotime) for expanding and further picking. They were picked and identified by PCR and western blot analysis. The validated clone was expanded with doxycycline and named *METTL3⁻/⁻*-METTL3-OE or *METTL3*-OE/KO hESCs. Similarly, *METTL14*-OE/KO hESCs were generated according this method. All gRNA sequences and primer sequences are listed in Supplementary Tables 1 and 4.

**Quantitative real-time PCR (qRT-PCR)**

We collected $5 \times 10^5$ hESCs and extracted total RNA from these cells using TRIzol (MRC). 2 μg of total RNA was respectively reverse transcribed with oligo dT or random primers in HiScript III 1st Strand cDNA Synthesis Kit (Vazyme, R312). Oligo dT primer was used for producing mature mRNA, and random primers were used for producing various RNA. We used chamQ SYBR qPCR Master Mix (Vazyme) and CFX96 machine (Bio-Rad) for qRT-PCR. We used GAPDH to normalize the human samples results and analyzed the data with three replicates. All primer sequences are listed in Supplementary Table 2.

**Western blot analysis**

The cells were collected and lysed with RIPA (Beyotime) containing cocktail (Roche) and PMSF (Sigma) on ice for 10 min. The whole cell extracts mixed with SDS loading buffer (Invitrogen) were boiled in 100 °C water for 10 min. The samples were loaded and run in 10% SDS-PAGE and transferred onto PVDF membranes (Millipore). These PVDF membranes were incubated with corresponding primary antibodies at 4 °C for 12 h. After washing three times with TBST for 10 min each time, the membranes were incubated with corresponding HRP-conjugated secondary antibodies at room temperature for 2 h. These membranes were washed three times in TBST for 10 min each time. The membranes were incubated and taken photos by ECL luminescent solution (TransGen Biotech) with an image analysis system (BLT PHOTON TECHNOLOGY), respectively. Detailed information about the antibodies used is listed in Supplementary Table 3 and original blots are provided in Source Data.

**Alkaline phosphatase (ALP) staining**

1000 single cells per well were plated on matrigel-coated 6-well plates in mTeSR1 plus 0.5 μM Thiazovivin for 1 day. Then the cells were cultured in mTeSR1 for 6 days. After 7 days, 4% paraformaldehyde was used to fix these cells at room temperature for 20 min. After washed three times with PBS, the cells were added with BCIP/NBT dyeing solution (Beyotime) at room temperature for 24 h. These samples were washed twice with ddH₂O and captured with a scanner.

**Teratoma formation analysis**

WT hESCs, *METTL3*-OE/KO hESCs, and *METTL14*-OE/KO hESCs were well maintained on matrigel-coated 6-well plates. Accuatse (Sigma) was used to disassociate these cells, and these cells were re-suspended with 30% matrigel in DMEM/F12 (Gibco). We injected these cells subcutaneously into immuno-deficient NOD-SCID mice at the age of about 4 weeks without DOX treatment[46]. Both male and female mice were used in teratoma formation experiment. 8 weeks later, these teratomas

were analyzed and fixed in 4% paraformaldehyde. Then these teratomas were stained with hematoxylin/eosin (H&E). The experiments involving animal research for teratoma formation had been reviewed and approved by IACUC at GIBH (NO. 2010012).

**Flow cytometry analysis**

The hES cells were digested as single cells by accutase and collected for the further procedures. Fixation buffer (BD Biosciences) was used to fix these cells at room temperature for 30 min. After washed once with PBS, these cells were permeated in perm/wash buffer (BD Biosciences) at 4 °C for 15 min. After that, these cells were incubated with corresponding primary antibodies at 37 °C for 30 min. After washed once with PBS, these cells were incubated with corresponding secondary antibodies at 37 °C for 30 min. These cells were washed once with PBS and re-suspended with PBS. Then, these samples were detected by Cytoflex (Beckman). Detailed information about the antibodies used is listed in Supplementary Table 3.

**Immuno-staining analysis**

The cells plated on matrigel-coated coverslips (NEST) were fixed in 4% paraformaldehyde at room temperature for 20 min. After washed thrice in PBS for 5 min each time, these samples were incubated with corresponding primary antibodies at 4 °C for 16 h. Then, after washed thrice in PBS for 5 min each time, these samples were co-incubated with corresponding secondary antibodies and DAPI (Sigma) at room temperature for 1.5 h. After these samples were washed thrice in PBS for 5 min each time, the coverslips were buckled back on the glass slides with fluorescence mounting medium (Dako) and were stored away from light at 4 °C. At least three fluorescence images per sample were analyzed by LSM 800 microscope (Zeiss). At least three fluorescence images of single cell per sample were captured by SP8-STED (Leica). Fluorescence intensity and fluorescence localization analysis were done by Image J. We used IMAGE J to convert the immunofluorescence pictures into image stacks for further analysis and then set the length of 8 μm in the nucleus. Moreover, we used IMAGE J to count the fluorescence values at 8 μm and form a corresponding fluorescence curve. Then, we chose the "Analyze" in GraphPad Prism to draw Smooth, differentiate, or integrate curve (parameter: Don't differentiate or integrate, and 20 neighbors on each size to average and 2nd order of the smoothing polynomial in Smooth).

Detailed information about the antibodies used is listed in Supplementary Table 3.

**PI cell cycle assay**

The cells were digested using accutase for 8 min and $5 \times 10^5$ cells were collected. These cells were washed once in PBS and fixed in 70% ethanol at 4 °C for 4 h. Then, these samples were incubated with PI staining solution (Beyotime) away from light at 37 °C for 0.5 h. These samples were analyzed with Cytoflex.

**EU assay**

The hESCs were disassociated using accutase for 8 min and $5 \times 10^5$ cells were collected. The cells were co-incubated with EU at 37 °C for 0.5 h and fixed with 4% paraformaldehyde at room temperature for 20 min. After

washed with PBS containing 3% BSA for 2 min, these cells were permeated with PBS containing 0.5% Triton-X-100 at room temperature for 10 min. After washed with PBS containing 3% BSA for 2 min, these samples were co-incubated with EU staining solution (ABP Biosciences) and DAPI at room temperature without light for 30 min. After washed with PBS for 2 min, these cells were re-suspended with PBS and detected by Cytoflex.

## RNA-seq analysis

$5 \times 10^5$ cells were collected and the total RNA was extracted with Trizol. Then, we used VAHTS mRNA-seq V3 Library Prep Kit for Illumina library kit (Vazyme) to generate sequencing libraries of transcriptome. The concentration of the libraries was detected by Qbuit (Thermo) and the fragment distribution of the libraries was detected by Q-sep100 (Bioptic). The sequencing libraries were sequenced by ANNOROAD Gene Technology to obtain the sequencing data. The original sequencing data were matched with the human (GRCh38/hg38) mRNA reference sequence by RSEM (rsem-1.2.4) and Bowtie2 (v2.2.5). The gene expression abundance was characterized by TPM (transcripts per million). The data were analyzed by glbase. The correlation of samples was analyzed by R software and the differentially expressed genes among samples were analyzed by the edgeR package. The heatmap analysis was performed using pheatmap.

## Cell membrane staining analysis

The cells plated on matrigel-coated coverslips (NEST) were fixed in 4% paraformaldehyde at room temperature for 20 min. After washed thrice in PBS for 5 min each time, these samples were incubated with 100 mM cell membrane dye DiD (KeyGEN, KGMP0025) and DAPI at 37 °C for 20 min. Then, after washed thrice in PBS for 10 min each time, these samples were buckled back on the glass slides with fluorescence mounting medium (Dako) and were stored away from light at 4 °C. At least three fluorescence images per sample were analyzed by LSM 800 microscope (Zeiss).

## Polyribosome analysis

In this experiment, the ribosomes in the cell lysate were separated with 10% to 50% sucrose gradient, and 10% and 50% sucrose solutions were prepared with ultra-pure water, respectively. After dissolving, 100 μg/mL Cycloheximide (CHX), 40 U/mL RNA enzyme inhibitor, and 1X protease inhibitor cocktail were added, and stored at 4 °C. The Beckman 13.2 mL ultra-clean tubes were prepared and BioComp Gradient Station automatic density gradient preparation instrument was used for density gradient preparation. Two plates of 10 cm plate for ribosome detection, and cell density is about 80%. Before collection, 100 μg/mL CHX should be added to the cell culture medium for 12 min, and then 0.25% trypsin was used to digest the cells for 3 min at 37 °C. After digestion, the samples were terminated using medium containing serum and 100 μg/mL CHX. After centrifugation, the supernatant was removed and the cells were collected. The collected cells were dissolved into cell lysate solution with 40 U/mL RNA enzyme inhibitors, 1X PMSF, and 100 μg/mL CHX at 15 min on ice. The samples were centrifuged at 4 °C for 15 min, and the total RNA concentration of the supernatants was detected (NanoDrop). The same total RNA of samples was added gently to the centrifuge tubes and were centrifuged at 4 °C, 180,000 × g for 2.5 h with SW-41 Ti (Beckman). After centrifugation, the samples were directly separated and analyzed by gradient with Biocomp automatic density gradient separation system. These data were analyzed by GraphPad Prism.

## Electron microscopic analysis

The hESCs plated on matrigel-coated coverslips (NEST) were fixed in 2.5% glutaraldehyde at room temperature for 15 min. After washed five times in PBS for 5 min each time, the samples were fixed in 1% osmic acid at room temperature for 20 min. After washed five times in PBS for 5 min each time, the samples were dehydrated with 50%, 70%, 80%, 90%, 100%, 100% ethanol separately. The samples were infiltrated with pure 812 resin for 4 h and in situ embedded. Then the samples were osmotic polymerized at 40 °C for 2 h and 60 °C for 12 h separately. After that, the samples were dyed with uranyl acetate and lead citrate. Electron microscopic images were analyzed by ultra-high resolution field emission frozen scanning electron microscope GeminiSEM 300 (Zeiss).

## Fluorescence recovery after photobleaching (FRAP)

Transfected NPM1-GFP hESCs plated on matrigel-coated coverslips (NEST) were maintained at 37 °C and 5% $CO_2$ in a humidity-controlled environment during acquisition. FRAP experiment was conducted using SP8-STED (Leica). The experiments were used argon ion laser. Firstly, we chose a fluorescent bleaching point. Before the experiments, three photos were taken and the fluorescence intensity values were recorded, and the average value of three fluorescence intensity values were taken. Then, we quenched it with 100% 488 nm laser and recorded fluorescence recovery times. For each FRAP experiment, the fluorescence was photobleached for 100 ms, and then a fixed confocal plane was acquired every 740 ms along a series of timelines during fluorescence recovery. Images were acquired using a frame size of $512 \times 512$ pixels. Additionally, these data were recorded and analyzed by GraphPad Prism.

## Co-immunoprecipitation

$1 \times 10^7$ cells were collected and dissolved into cell lysate (50 mM Tris-HCl PH 7.6, 150 mM KCl, 1% Triton-X-100, 1 mM EDTA, 10% glycerol) containing cocktail and PMSF on ice for 30 min. After these samples were centrifuged at 4 °C for 20 min, the supernatant of these samples was collected as the whole cell extract. 60 μL of whole cell extract was taken as Input. Protein A (Invitrogen) and protein G (Invitrogen) magnetic beads were washed thrice in IP washing buffer (50 mM Tris-HCl PH 7.6, 150 mM KCl, 0.1% Triton-X-100, 1 mM EDTA, 10% glycerol) containing cocktail and PMSF at 4 °C for 5 min each time. 100 μL whole cell extract, 200 μL IP buffer (50 mM Tris-HCl PH 7.6, 150 mM KCl, 1 mM EDTA, 10% glycerol) containing cocktail and PMSF, and 2 μL corresponding antibodies were co-incubated with Protein A and protein G at 4 °C for 16 h. Then, these samples were washed four times in IP washing buffer containing cocktail and PMSF for 5 min each time. These samples were put on the magnetic frame to remove the IP wash buffer completely. The samples were eluted with 60 μL 1x SDS loading buffer and boiled for 10 min. These samples were analyzed by Western blotting. Detailed information about the antibodies used is listed in Supplementary Table 3.

## CUT&TAG sequencing

CUT&TAG-seq libraries of corresponding samples were performed using the Hyperactive Universal CUT&Tag Assay Kit for Illumina Pro (Vazyme, TD904) according the manufacturer's recommendations and were run on an Illumina NovaSeq 6000 platform. The adapters of raw fastq files were cut using Cutadapt (v1.12) with parameters -m 18 -q 30,30 --max-n = 0.05 -e 0.2 -n 2. Then, bowtie2 (v2.2.9) was used to align above processed sequences to the reference human genome (hg38) with parameters -p 8 --very-sensitive-local --no-unal --no-mixed --no-discordant -- phred33 -I 10 -X 700 -x spike_in_genome.fa. SAMtools (v1.3) was used for format conversion and picard-MarkDuplicates (v1.119) was used to tag duplicated reads. MACS2 (v2.2.6) was used for peak calling of H3K9me3 with parameters -t sample1.bam -g hs -f BAMPE --broad -n sample1 --keep-dup=all --ourdir. DeepTools (v3.5.1) was used to generate the bigwig signal files and then bamCoverage, computeMatrix, and plotProfile modules in DeepTools (v3.5.1) were used for visualizing them. The annotatePeak function of R package ChIPseeker (v1.38.0) was used for peak annotation. ClusterProfiler (v3.18.0) of R package was used for Gene Ontology (GO) enrichment analysis.

**Fluorescence resonance energy transfer (FRET) measurement**

The samples were processed following immunostaining method and the corresponding images were analyzed by the model FRET-AB of Leica SP8 STED. The FRET pair in this manuscript was Alexa Fluor 488 (donor) and Alexa Fluor 568 (acceptor). The FRET efficiency was calculated with parameter "FRETeff = (post$^{acceptor}$–pre$^{acceptor}$)/post$^{acceptor}$", and the fluorescence intensity of the acceptor before photobleaching and after photobleaching is post$^{acceptor}$ and pre$^{acceptor}$, respectively.

**Statistics and reproducibility**

In general, the data were presented as the mean ± SD (standard deviation) from at least three independent repeats. We used unpaired two-tailed Student's tests ($t$ test) to determine the significance level and considered $P$ value < 0.05 as the statistically significant difference. No samples were excluded from any analysis. Additionally, we presented the representative morphology, immune-staining pictures, and western blot results including Figs. 1a, b, 4d, e, g, and 5b–f from at least three replicates.

**Reporting summary**

Further information on research design is available in the Nature Portfolio Reporting Summary linked to this article.

## Data availability

The raw data of RNA sequencing and CUT&TAG sequencing generated in this study have been deposited in the Genome Sequence Archive under the accession code HRA002951 and HRA007207. qRT-PCR data, original western blots, the quantification results of FACS, corresponding western blot, and fluorescence intensity have also been deposited in Figshare [https://doi.org/10.6084/m9.figshare.25623738] and Source Data file. Source data are provided with this paper.

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

## Acknowledgements

We thank the lab members in GIBH for their kind help. This work was supported by the National Key Research and Development Program of China, Stem Cell and Translational Research (2022YFA1105001 to G.P.); Research Funds from Health@InnoHK Program launched by Innovation Technology Commission of the Hong Kong SAR, P. R. China (to G.P.); the National Natural Science Foundation of China (32270624 to Y.S., 31971374 to G.P.); the Youth Innovation Promotion Association of the Chinese Academy of Sciences (2022360 to Y.S.); China Postdoctoral Science Foundation Funded Project (2023M733516 to Y.Z.); Science and Technology Planning Project of Guangdong Province, China (2023B1212060050 to G.P., 2023B1212120009 to G.P.); Fountain-Valley Life Sciences Fund of University of Chinese Academy of Sciences Education Foundation (ZXXM202201 to G.P.); Guangzhou Key Research and Development Program (202206010041 to G.P.); Guangdong Provincial Key Laboratory of Stem Cell and Regenerative Medicine (2020B1212060052 to G.P.); the Guangdong Province Special Program for Outstanding Talents (2019JC05Y463 to G.P.).

## Author contributions

G.P. and Y.S. designed and directed the project, G.P., Y.S. and Y.Z. co-wrote the manuscript. Y.S. Y.Z. and Y.W. carried out most of the experiments and analyzed result data. Y.S., Y.Z. and C.Z. carried out gene editing for METTL3/METTL14 knock-out and knock-in in hESCs. Y.S. and Y.Z. carried out WB experiments. Q.X. performed polysome profiling experiment. Y.Z. and W.G. performed immuno-staining analysis and FRAP analysis. H.Li performed electron microscopic analysis. J.W. and Y.Z. performed FACS and RT-qPCR. T.Z. carried out teratoma and kar-yotype analysis. Q.C., Y.L., H.Lin and J.H. carried out RNA-seq and ana-lyzed the RNA-seq data. J.C. gave suggestions about experiments. All authors read and approved the final manuscript.

## Competing interests

The authors declare no competing interests.
