## [Peer Review File · Nature Communications]

METTL3/METTL14 maintain human nucleoli integrity by mediating SUV39H1/H2 degradationREVIEWER COMMENTS

Reviewer #1 (Remarks to the Author):

This manuscript by Shan et al. aims to show that the m6A methyltransferase complex of METTL3/METTL14 helps maintain nucleoli integrity by acting as a scaffold between the H3K9 methyltransferase SUV39H1/H2 and the E3 ubiquitin ligase CRL4. First, they show that loss of either METTL3 or METTL14 disrupts nucleolar structure/function and impairs the ability of human embryonic stem cells (hESCs) to self-renew. Next, they show that loss of METTL3 or METTL14 leads to an increase in protein levels of SUV39H1/H2 and H3K9 methylation in the nucleolus, which impairs liquid-liquid phase separation (LLPS). In addition, they show that the nucleolar defects seen with loss of METTL3 or METTL14 can be rescued by catalytically active METTL3 or METTL14 but not by catalytically inactive mutants. Finally, they show that METTL3 interacts with CRL4 and that this is dependent on the production of nascent pre-RNAs. They propose a model where, instead of methylating SUV39H1/H2 RNA, METTL3/METTL14 regulates SUV39H1/H2 protein levels by providing a scaffold to CRL4 (in the presence of nascent RNA chains) and promoting its ubiquitination and degradation.

This work describes a novel role for METTL3 in the nucleolus, outside of its traditional role as an m6A methyltransferase. I believe it is suitable for publication in Nature Communications, if the following points are addressed.

Major points:

- 1.) A large focus of this paper is that "...METTL3/METTL14 maintain nucleolar integrity depending on their RNA methyltransferase activity." (lines 282-283). However, their model posits that m6A methylation is not needed, but in the nucleolus METTL3/METTL14 is actually acting as a scaffold. Their model speaks to its RNA-binding ability, rather than its m6A activity as necessary. While the authors state that the mutants used are catalytically inactive, no detail is given (only a citation). More explanation should be given on these mutations and whether they actually disrupt RNA-binding and/or just catalytic activity. As METTL14 is not actually a catalytically active enzyme, I think clarifying this discrepancy will go a long way to strengthening this model.
- 2.) Another focus of the paper is both METTL3 and METTL14 are needed for this function in the nucleolus. However, for the experiments that show that in the nucleolus they are not acting as a methyltransferase complex, but a scaffold to CRL4 (Fig. 5), only METTL3 is tested. For functions that don't involve its m6A methylation activity, METTL3 has been shown to work independently from METTL14. Therefore, it would be interesting to see if METTL14 is actually in the complex with CRL4 and METTL3. This could be done by testing the DDB1 IPs for METTL14.

Minor points

- 1.) Why is there no METTL14 in METTL3 knockdown? (Fig. 1A) Does this mean you cannot interpret any results as only METTL3 loss?
- 2.) Western blots used to compare expression or pulldown should be quantitated (as was done for Fig. 1i), making sure to ratio pulldown to expression levels. This includes Fig. 3a,c and Fig. 5b,c,e,f. Though some experiments are clear by eye, without quantification, you cannot tell if the data is

reproducible.

3.) Data saying m6A methylation is in the nucleoli is very weak (Fig. 4ab). The staining patterns are very different with very little overlap. Since no claim is made that m6A methylation is needed for function in the nucleolus, the point of this weak data is unclear. I would suggest removing it.

4.) Figure 4E only pSIN-WT-METTL3 rescues, not WT METTL14. What is concluded from this?

5.) Figure 4F, the claim cannot be made that overexpressed WT METTL3 localized to the nucleoli, as it has very high expression across the whole diameter of the cell that does not have much fluctuation. Also can't say the mutants reduced nucleolar localization, they also have diffuse localization, just to a much lesser degree.

6.) Line 190 supplementary figure number is missing.

Reviewer #2 (Remarks to the Author):

In this study, Shan et al. report a function of the m6A METTL3/METTL14 methyltransferase complex, in regulation of SUV39H1/2 and heterochromatin degradation in human embryonic stem cells (hESCs). Apart from its more established post-transcriptional effects on mRNA stability and translation, m6A has been reported to regulate transcription, premature termination and heterochromatin. The present study uncovers requirements for METTL3 and METTL14 in maintaining self-renewal capacity of hESCs, which correlate with degradation of H3K9me3 methyltransferase SUV39H1/2. They found METTL3/METTL14 serves as an essential adaptor for CRL4 E3 ubiquitin ligase targeting SUV39H1/H2 for polyubiquitination and proteasomal degradation and therefore prevents H3K9me3 accumulation in nucleoli. In addition, this regulation is dependent on m6A. I think this finding is very interesting. I hope the authors could do some experiments to make their conclusions solidier, especially this regulation is depending on m6A.

1: Line 190-194, If the authors want to declare that m6A is necessary for hESCs self-renewal, I would be helpful for the authors to detect m6A levels in different cell. More importantly, METTL14's EPPL motif has been reported to lost activity and METTL14 has been reported to be as co-factor for METTL3's activity (ref 32 in this manuscript), it seems un-reasonable to use METTL14 APPA mutation as control. METTL14 R298 and D312 mutations have been reported to abolish METTL3-mediated m6A methylation (ref 32 in this manuscript). If the authors want to declare m6A is necessary, the authors should rescue METTL14 R298 and D312 mutations, not APPA mutation in METTL14 deletion cells.

2: Are other writers required (WTAP, KIAA1429...)? The authors could knock out m6A demethylase ALKBH5 or FTO to see if the SUV39H1/2 and heterochromatin decreased?

3: If this regulation is dependent on m6A, I believe this would be m6A reader(s) involved. Generally, YTHDC1 is reported to be in nucleus. It would be great for the authors to knock out YTHDC1 to check if the similar phenotype could be observed.

4: METTL3/METTL14 promotes the degradation of H3K9me3 methyltransferase SUV39H1/2. Are

these regulations only on H1 hESCs, or is a general regulation? It would be great if the authors could check this in mouse embryonic stem cells and others cells (Such as HEK293T and HeLa cells).

5: Figure 1c, the down-regulated genes in METTL3 and METTL14 deletion hESCs were enriched in functions related to cellular metabolism, translation, rRNA processing as well as cell cycle progression etc. Are H3K9me3 specifically increased in these genes? It would be helpful if the authors could perform SUV39H1/2 and H3K9me3 ChIP experiments.

Reviewer #3 (Remarks to the Author):

In the manuscript by Shan et al, the authors focused on a particular functional link between the m6A RNA complex and nucleolar structure and content. They presented in their data that METTL3/METTL14 deficiency affects the self-renewal of human embryonic stem cells and suggest that this is mediated by changes in nucleolar integrity. The authors proposed that the METTL3/METTL14 complex acts through methylation of m6A rRNA in the nucleolus to prevent accumulation of H3K9me3 by proteasomal degradation of the H3K9 methyltransferase SUV39H1/2, enabling correct liquid-liquid phase separation (LLPS). In addition, they provide mechanical linkage through METTL3/14 recruitment of the Cullin 4-RING ligase complex (CRL4) to promote SUV39H1/2 ubiquitylation and degradation. Although the study of the involvement of the m6A machinery in nucleolar integrity is highly original and once again bridges epitranscriptomics and epigenetics in a novel context, I have a number of concerns about the consistency of the results with the working model. Finally, as other m6A methyltransferases ZCCHC4 and METTL5 (Van Tran N et al., 2019, Ma H et al., 2019) have been shown to be involved in rRNA methylation, it would be important to discuss these findings and how they may fit into the model proposed in this manuscript.

Major concerns:

First, I have a concern related to a technical aspect. Throughout the paper, the authors show measurements of signal intensity in different contexts and using several markers. What I find a little problematic is that in many cases, the authors have drawn conclusions on fluorescence histograms along the arbitrary virtual section of one nucleolus. I think that presenting an average quantification with SD of several sections and several nucleoli per condition would allow a more robust interpretation of the results. Furthermore, in many cases (Fig2b, Fig3, Fig4 f-k), the fluorescent signal appears saturated in METTL3/14 deficiency, which may lead to misinterpretation. It would be needed to have a detailed description in mat & met on how these pictures have been processed. Moreover, as nucleoli are smaller in KO conditions, it is probably necessary to normalize them with respect to their length.

My second major concern relates to the nucleolus specificity of the model and the link with rRNA. The authors clearly indicate an increase in H3K9me3 in nucleoli that correlates with SUV39H1/2 in METTL3/14 deficiency. However, we can also observe an overall nuclear increase which should also be quantified and which opens up questions (see below). Another problem is that METTL3/14 does not seem to accumulate in the nucleolus, which explains

why there is such an increase in SUV39H1/2 and H3K9me3 in particular in the nucleolus. rRNA is very abundant, so if the model is correct, we should see an increase in METTL3/14. It would be important to determine more clearly that nascent rRNA is the recruiting factor for the m6A complex and that this is the starting point for SUV39H1/2 degradation in the nucleolus. The use of CasRX-Alkbh5 vs METTL3 specifically targeting pre-rRNA could be an option to demonstrate this.

It would also be interesting to test whether a gain of function of SUV39H1/2 and not SETB1 is sufficient to reproduce nucleolar integrity defects.

The authors have used Cpo-IP (in Fig5e,f) to show interactions between METTL3/14 with SUV39H1/2 and the CUL4A/B complex, but this does not show that this occurs in the nucleolus. It would be important to show these interaction inside the nucleolus, perhaps by PLA experiments.

Third, as the level of H3K9me3 increases strongly in the nucleolus but also in nuclear periphery (and also that the nucleus shape seems different) after METTL3/14 deficiency it may indicate that chromatin organization is affected. It would be interesting to check the global changes occurring on H3K9me3 occupancy by ChIP-seq.

We can't also rule out considering the pleiotropic function of m6A machinery that a global cellular stress is occurring after KO leading to additional nucleoli instability. This should be discuss by authors also it is not clear whether METTL3/14 KO impact the level and/or function of proteins participating in nucleoli LLPS: NPM1, FBL..

This is important to check. Also it would be nice to verify if there is any impact on ribosomal subunits synthesis in KO which may account for decrease in polysome profiling.

Minor concerns:

It is not clear from the study whether nucleoli are analyzed in the context of polynucleated or mononucleated cells in METTL3/14 KO.

Surprisingly, the METTL14 protein appears to be completely lost in METTL3 KO. Does this mean that the METTL3 KO condition is in fact a double KO?

I'm also not very convinced, based on the images available, that the nucleoli of mutant hESCs have very different shapes to those of WT hESCs.

Fig2: The top panel d (FRAP experiment) is not very visible. I suggest removing it or improving the quality.

Fig3a: Why is there no detection of H3K9me3 in the WT condition?

Fig3d: Isn't it surprising that the intensity of the Pyronin signal is stronger in the mutant context when we would expect to see less rRNA in the nucleoli?

Fig4c: it's hard to see anything on this panel.

Fig4e and Sup Fig5 d: the color code for each condition should be the same.

Fig4d-g: there is a lack of consistency between the western blot and the fluorescence measurement, particularly for the mutant condition. The fluorescent signal is much weaker in the mutant condition than in the WT condition, so it's difficult to draw any conclusions from this experiment.

Fig5d: I think the anti-myc-Ub blot is missing to see where the 55KD and 40KD scale bands are.

Fig5e: It would have been interesting to add an RNase treatment to see if these interactions are RNA-dependent.

REVIEWER COMMENTS

Reviewer #1 (Remarks to the Author):

This manuscript by Shan et al. aims to show that the m⁶A methyltransferase complex of METTL3/METTL14 helps maintain nucleoli integrity by acting as a scaffold between the H3K9 methyltransferase SUV39H1/H2 and the E3 ubiquitin ligase CRL4. First, they show that loss of either METTL3 or METTL14 disrupts nucleolar structure/function and impairs the ability of human embryonic stem cells (hESCs) to self-renew. Next, they show that loss of METTL3 or METTL14 leads to an increase in protein levels of SUV39H1/H2 and H3K9 methylation in the nucleolus, which impairs liquid-liquid phase separation (LLPS). In addition, they show that the nucleolar defects seen with loss of METTL3 or METTL14 can be rescued by catalytically active METTL3 or METTL14 but not by catalytically inactive mutants. Finally, they show that METTL3 interacts with CRL4 and that this is dependent on the production of nascent pre-RNAs. They propose a model where, instead of methylating SUV39H1/H2 RNA, METTL3/METTL14 regulates SUV39H1/H2 protein levels by providing a scaffold to CRL4 (in the presence of nascent RNA chains) and promoting its ubiquitination and degradation.

This work describes a novel role for METTL3 in the nucleolus, outside of its traditional role as an m⁶A methyltransferase. I believe it is suitable for publication in Nature Communications, if the following points are addressed.

Response: We thank the reviewer for the positive comments.

Major points:

1.) A large focus of this paper is that "...METTL3/METTL14 maintain nucleolar integrity depending on their RNA methyltransferase activity." (lines 282-283). However, their model posits that m⁶A methylation is not needed, but in the nucleolus METTL3/METTL14 is actually acting as a scaffold. Their model speaks to its RNA-binding ability, rather than its m⁶A activity as necessary. While the authors state that the mutants used are catalytically inactive, no detail is given (only a citation). More explanation should be given on these mutations and whether they actually disrupt RNA-binding and/or just catalytic activity. As METTL14 is not actually a catalytically active enzyme, I think clarifying this discrepancy will go a long way to strengthening this model.

Response: Thanks for this suggestion. We agree with this comment and revised our manuscript based on the reviewer's suggestion. The mutation sites used here are reported critical motifs for METTL3 or 14 function. We did not re-perform experiments to examine these mutant forms in detail because these experiments are not the main focus in current manuscript, thus we revised text to be more appropriate and focus more on their RNA binding rather than m⁶A methylation. Indeed, our new experiments that after RNase treatment to destroy the RNA, the interactions of METTL3, METTL14, DDB1 and SUV39H1/2 were impaired (Supplementary Fig. 8a), suggesting that RNA binding is critical in this process. We revised the model and text accordingly.

2.) Another focus of the paper is both METTL3 and METTL14 are needed for this function in the nucleolus. However, for the experiments that show that in the nucleolus they are not acting as a methyltransferase complex, but a scaffold to CRL4 (Fig. 5), only METTL3 is tested. For functions that don't involve its m⁶A methylation activity, METTL3 has been shown to work independently from METTL14. Therefore, it would be interesting to see if METTL14 is actually in the complex with CRL4 and METTL3. This could be done by testing the DDB1 IPs for METTL14.

Response: Thanks for this suggestion. We performed the DDB1 IPs for METTL14 and showed that METTL14 interacted with CRL4 and METTL3 (Fig. 5e).

Minor points

1.) Why is there no METTL14 in METTL3 knockdown? (Fig. 1A) Does this mean you cannot interpret any results as only METTL3 loss?

Response: Thanks for this suggestion. This is an interesting phenomenon. In previous report in mouse ESCs, Mettl3 knockdown also resulted in Mettl14 loss (ref 30 and 31 in the manuscript). This phenotype indicates that integrity of the complex is critical for the stability of METTL14 protein.

2.) Western blots used to compare expression or pulldown should be quantitated (as was done for Fig. 1i), making sure to ratio pulldown to expression levels. This includes Fig. 3a, c and Fig. 5b, c, e, f. Though some experiments are clear by eye, without quantification, you cannot tell if the data is reproducible.

Response: Thanks for this suggestion. We performed the more western blots and provide quantitative data for these figures.

3.) Data saying m⁶A methylation is in the nucleoli is very weak (Fig. 4ab). The staining patterns are very different with very little overlap. Since no claim is made that m⁶A methylation is needed for function in the nucleolus, the point of this weak data is unclear. I would suggest removing it.

Response: Thanks for this suggestion. We removed these results according to this suggestion. Please see the new figure 4.

4.) Figure 4E only pSIN-WT-METTL3 rescues, not WT METTL14. What is concluded from this?

Response: Thanks for this suggestion. This result means that METTL3 and METTL14 are not functionally redundant. We agree that this result is not very useful here, so we just remove it to avoid misleading.

5.) Figure 4F, the claim cannot be made that overexpressed WT METTL3 localized to the nucleoli, as it has very high expression across the whole diameter of the cell that does not have much fluctuation. Also can't say the mutants reduced nucleolar localization, they also have diffuse localization, just to a much lesser degree.

Response: Thanks for this suggestion. We agree the comment that the over-expression might lead an improper sub-cellular localization and we remove these results because it might not be critical in this context.

6.) Line 190 supplementary figure number is missing.

Response: Thanks for this suggestion. We revised the manuscript.

Reviewer #2 (Remarks to the Author):

In this study, Shan et al. report a function of the m⁶A METTL3/METTL14 methyltransferase complex, in regulation of SUV39H1/2 and heterochromatin degradation in human embryonic stem cells (hESCs). Apart from its more established post-transcriptional effects on mRNA stability and translation, m⁶A has been reported to regulate transcription, premature termination and heterochromatin. The present study uncovers requirements for METTL3 and METTL14 in maintaining self-renewal capacity of hESCs, which correlate with degradation of H3K9me3 methyltransferase SUV39H1/2. They found METTL3/METTL14 serves as an essential adaptor for CRL4 E3 ubiquitin ligase targeting SUV39H1/H2 for polyubiquitination and proteasomal degradation and therefore prevents H3K9me3 accumulation in nucleoli. In addition, this regulation is dependent on m⁶A. I think this finding is very interesting. I hope the authors could do some experiments to make their conclusions solidier, especially this regulation is depending on m⁶A.

Response: We thank the reviewer for the positive comments.

1: Line 190-194, If the authors want to declare that m⁶A is necessary for hESCs self-renewal, I would be helpful for the authors to detect m⁶A levels in different cell. More importantly, METTL14's EPPL motif has been reported to lost activity and METTL14 has been reported to be as co-factor for METTL3's activity (ref 32 in this manuscript), it seems un-reasonable to use METTL14 APPA mutation as control. METTL14 R298 and D312 mutations have been reported to abolish METTL3-mediated m⁶A methylation (ref 32 in this manuscript). If the authors want to declare m⁶A is necessary, the authors should rescue METTL14 R298 and D312 mutations, not APPA mutation in METTL14 deletion cells.

Response: Thanks for this suggestion. As pointed by this reviewer and also the reviewer 1, the main finding in current manuscript is uncovering a non-conventional role of METTL3/14 as an adaptor for SUV39H1/2 degradation and thus nucleoli integrity. As we responded to reviewer 1, the motifs selected for mutation were reported to be critical motifs for METTL3/14 functions, not only catalytic activity. However, we did not examine these mutations in detail in current manuscript since they fell outside the main topic in current manuscript. We revised the text to be more appropriate. However, our new experiments that after RNase treatment, the interactions of METTL3, METTL14, DDB1 and SUV39H1/2 were impaired (Supplementary Fig. 8a), suggesting that RNA binding is critical in this process. We revised the model and text accordingly.

2: Are other writers required (WTAP, KIAA1429...)? The authors could knock out m6A demethylase ALKBH5 or FTO to see if the SUV39H1/2 and heterochromatin decreased?

Response: Thanks for this suggestion. We haven't done these mutants yet but it's definitely worth to pursue further. We discussed it in the text.

3: If this regulation is dependent on m⁶A, I believe this would be m6A reader(s) involved. Generally, YTHDC1 is reported to be in nucleus. It would be great for the authors to knock out YTHDC1 to check if the similar phenotype could be observed.

Response: Thanks for this suggestion. We did perform YTHDC1 deletion in human ESCs (*YTHDC1*^{-/-}). However, *YTHDC1*^{-/-} hESCs displayed undifferentiated phenotype, in big contrast to *METTL3*^{-/-} and *METTL14*^{-/-} hESCs (see below). These data indicate a different role of YTHDC1 in hESCs, which is another research project to further pursue. We just discussed these results in the text and did not include them in current manuscript.

(a) Morphology of wild type (WT) and *YTHDC1*^{-/-} hESCs. Scale bar, 50 μ m. (b) western blot of YTHDC1 or H3K9me3 in indicated cells. (c-d) Immunostaining on the nucleolus marker NPM1 and the quantification of nucleolus number in the indicated cell lines, respectively. Scale bar, 10 μ m. (e) Immunostaining on FBL (DFC marker), NPM1 (GC marker) in WT and *YTHDC1*-deficient hESCs. Scale bar, 3 μ m. (f) Immunostaining on H3K9me3 and nucleolar marker NPM1 in the indicated cells. Scale bar, 3 μ m.

4: METTL3/METTL14 promotes the degradation of H3K9me3 methyltransferase SUV39H1/2. Are these regulations only on H1 hESCs, or is a general regulation? It would be great if the authors could check this in mouse embryonic stem cells and others cells (Such as HEK293T and HeLa cells).

Response: Thanks for this suggestion. We did perform knock-down of METLL3 in 293T and HeLa cells, but did not see a significant phenotype changes similar to hESCs (see below). In previous reports, the deletion of Mettl3 actually results in upregulation of H3K9me3 enrichment on chromatin (Ref 26 and 27 in the manuscript). We mentioned this in the text, but did not include the data in current manuscript since they might not be directly relevant.

(a-b) Western blot of H3K9me3 and its methyltransferases SUV39H1/2 in METTL3-knockdown 293T and HeLa cells.

5: Figure 1c, the down-regulated genes in METTL3 and METTL14 deletion hESCs were enriched in functions related to cellular metabolism, translation, rRNA processing as well as cell cycle progression etc. Are H3K9me3 specifically increased in these genes? It would be helpful if the authors could perform SUV39H1/2 and H3K9me3 ChIP experiments.

Response: Thanks for this suggestion. We performed CUT & TAG for H3K9me3 enrichment on chromatin and found that the H3K9me3 enrichment was increased globally consistent to significant upregulation of H3K9me3 proteins (Supplementary Fig. 5f-i). Indeed, H3K9me3 enriched genes include genes related to cellular metabolism, translation, rRNA processing as well as cell cycle progression was increased (Supplementary Fig. 5i).

Reviewer #3 (Remarks to the Author):

In the manuscript by Shan et al, the authors focused on a particular functional link between the m6A RNA complex and nucleolar structure and content. They presented in their data that METTL3/METTL14 deficiency affects the self-renewal of human embryonic stem cells and suggest that this is mediated by changes in nucleolar integrity. The authors proposed that the METTL3/METTL14 complex acts through methylation of m6A rRNA in the nucleolus to prevent accumulation of H3K9me3 by proteasomal degradation of the H3K9 methyltransferase SUV39H1/2, enabling correct liquid-liquid phase separation (LLPS). In addition, they provide mechanical linkage through METTL3/14 recruitment of the Cullin 4-RING ligase complex (CRL4) to promote SUV39H1/2 ubiquitylation and degradation. Although the study of the involvement of the m6A machinery in nucleolar integrity is highly original and once again bridges epitranscriptomics and epigenetics in a novel context, I have a number of concerns about the consistency of the results with the working model. Finally, as other m6A methyltransferases ZCCHC4 and METTL5 (Van Tran N et al., 2019, Ma H et al., 2019) have been shown to be involved in rRNA methylation, it would be important to discuss these findings and how they may fit into the model proposed in this manuscript.

Response: We thank the reviewer for the positive comments. We did perform more experiments and revised our manuscript accordingly.

Major concerns:

First, I have a concern related to a technical aspect. Throughout the paper, the authors show measurements of signal intensity in different contexts and using several markers. What I find a little problematic is that in many cases, the authors have drawn conclusions on fluorescence histograms along the arbitrary virtual section of one nucleolus. I think that presenting an average quantification with SD of several sections and several nucleoli per condition would allow a more robust interpretation of the results. Furthermore, in many cases (Fig2b, Fig3, Fig4 f-k), the fluorescent signal appears saturated in METTL3/14 deficiency, which may lead to misinterpretation. It would be needed to have a detailed description in mat & met on how these pictures have been processed. Moreover, as nucleoli are smaller in KO conditions, it is probably necessary to normalize them with respect to their length.

Response: Thanks for this suggestion. In revised version, we provided five representative nucleoli per condition and quantitative data on fluorescence intensity. We also provided detailed description on how the image was processed in Methods and Materials.

My second major concern relates to the nucleolus specificity of the model and the link with rRNA. The authors clearly indicate an increase in H3K9me3 in nucleoli that correlates with SUV39H1/2 in METTL3/14 deficiency. However, we can also observe an overall nuclear increase which should also be quantified and which opens up questions (see below).

Response: Thanks for this suggestion. We re-analyzed these results more quantitatively. Indeed, H3K9me3 and SUV39H1/2 were increased both in nucleoli and an overall nucleus, however, the signal increased a lot more in nucleoli compared with other regions in nucleus (new Fig. 3b-e). We discussed this in the text.

Another problem is that METTL3/14 does not seem to accumulate in the nucleolus, which explains why there is such an increase in SUV39H1/2 and H3K9me3 in particular in the nucleolus. rRNA is very abundant, so if the model is correct, we should see an increase in METTL3/14.

Response: Thanks for this comment. Based on our immunostaining results, METTL3 did show localization on nucleolus. To further solidate this finding, we provided new analysis that METTL3 contained potential nucleolar localization signal sequence (NoLS). Mutation of the motif (METTL3^{ΔNoLS}) clearly impaired its nucleolar localization and did not rescue nucleolar functions (new Fig. 4e-g). These data indicate that nucleolar localization of METTL3 might not be much more abundant compared with other regions in nuclei, but is critical for nucleolar functions. We discussed this in the text.

It would be important to determine more clearly that nascent rRNA is the recruiting factor for the m⁶A complex and that this is the starting point for SUV39H1/2 degradation in the nucleolus. The use of CasRX-Alkbh5 vs METTL3 specifically targeting pre-rRNA could be an option to demonstrate this.

Response: Thanks for this suggestion. Instead of the suggested experiments, we performed co-IP assay by anti-METTL3 antibody with RNase A treatment and showed that the interactions of METTL3 with DDB1 and SUV39H1/2 were greatly decreased, suggesting that these interactions are RNA-dependent (Supplementary Fig. 8a).

It would also be interesting to test whether a gain of function of SUV39H1/2 and not SETB1 is sufficient to reproduce nucleolar integrity defects.

Response: Thanks for this suggestion. We performed overexpression of SUV39H1/2 and SETDB1 in wild type hESCs and showed that only SUV39H1/2 and not SETDB1 were sufficient to reproduce nucleolar integrity defects (Supplementary Fig. 5a-e).

The authors have used Co-IP (in Fig5e, f) to show interactions between METTL3/14 with SUV39H1/2 and the CUL4A/B complex, but this does not show that this occurs in the nucleolus. It would be important to show these interactions inside the nucleolus, perhaps by PLA experiments.

Response: Thanks for this suggestion. Instead of suggested PLA experiments that is not available in the lab, we firstly performed FRET assay to show that the interactions between METTL3 with SUV39H1/2 and DDB1 occurred in nucleus (Supplementary Fig. 7e-g). Further, we performed co staining of DDB1 and METTL3 and showed that they co-localized within nucleolus (Supplementary Fig. 7g).

Third, as the level of H3K9me3 increases strongly in the nucleolus but also in nuclear periphery (and also that the nucleus shape seems different) after METTL3/14 deficiency it may indicate that chromatin organization is affected. It would be interesting to check the global changes occurring on H3K9me3 occupancy by ChIP-seq.

Response: Thanks for this suggestion. We performed CUT&TAG for detecting H3K9me3 enrichment on chromatin. We did find a global increase of H3K9me3 enrichment after METTL3 deficiency (Supplementary Fig. 5f-i).

We can't also rule out considering the pleiotropic function of m6A machinery that a global cellular stress is occurring after KO leading to additional nucleoli instability. This should be discussed by authors also it is not clear whether METTL3/14 KO impact the level and/or function of proteins participating in nucleoli LLPS: NPM1, FBL.. This is important to check. Also it would be nice to verify if there is any impact on ribosomal subunits synthesis in KO which may account for decrease in polysome profiling.

Response: Thanks for this suggestion. Based on RNA-seq data, we found that the expression level of NPM1, FBL and UBF in METTL3/14-KO hESCs was similar with that of wild type hESCs (Supplementary Fig. 1j). However, heatmap analysis showed down-regulation of genes for ribosomal subunits in METTL3/14-KO hESCs (Supplementary Fig. 3d). We discussed this accordingly.

Minor concerns:

It is not clear from the study whether nucleoli are analyzed in the context of polynucleated or mononucleated cells in METTL3/14 KO.

Response: Thanks for this suggestion. We analyzed nucleoli in the context of polynucleated cells in METTL3/14-KO hESCs. For example, nucleolar number was analyzed in all DAPI-staining cells in the pictures as Fig. 1e, and nucleolar size/diameter and fluorescence intensity of the indicated antibodies were analyzed in random five DAPI-staining cells.

Surprisingly, the METTL14 protein appears to be completely lost in METTL3 KO. Does this mean that the METTL3 KO condition is in fact a double KO?

Response: Thanks for this suggestion. This is an interesting phenomenon. In previous report in mouse ESCs, Mettl3 knockdown also resulted in Mettl14 loss (ref 30 and 31 in the manuscript). This phenotype indicates that integrity of the complex is critical for the stability of METTL14 protein.

I'm also not very convinced, based on the images available, that the nucleoli of mutant hESCs have very different shapes to those of WT hESCs.

Response: Thanks for this suggestion. We agree with the comment and revised the text.

Fig2: The top panel d (FRAP experiment) is not very visible. I suggest removing it or improving the quality.

Response: Thanks for this suggestion. We removed these results accordingly.

Fig3a: Why is there no detection of H3K9me3 in the WT condition?

Response: Thanks for this suggestion. H3K9me3 was substantially increased in METTL3/14-KO hESCs but was very low in WT cells. We performed more western blots and quantitated data for a stronger conclusion (Fig. 3a).

Fig3d: Isn't it surprising that the intensity of the Pyronin signal is stronger in the mutant context when we would expect to see less rRNA in the nucleoli?

Response: Thanks for this suggestion. Pyronin is a cationic dye that intercalates total RNA, not only rRNA. We performed RNA electrophoresis with total RNA from 0.5 million cells and showed the less rRNA in METTL3/14-KO hESCs, consistent with qRT-PCR result (Supplementary Fig. S1h, Fig. 1g).

Fig4c: it's hard to see anything on this panel.

Response: Thanks for this suggestion. This result means that METTL3 and METTL14 are not functionally redundant. We agree that this result about METTL14 is not very useful here, so we removed it to avoid misleading.

Fig4e and Sup Fig5 d: the color code for each condition should be the same.

Response: Thanks for this suggestion. We revised color code accordingly.

Fig4d-g: there is a lack of consistency between the western blot and the fluorescence measurement, particularly for the mutant condition. The fluorescent signal is much weaker in the mutant condition than in the WT condition, so it's difficult to draw any conclusions from this experiment.

Response: Thanks for this suggestion. We agree the comment that the over-expression might lead an improper localization, and we removed these results about fluorescent signal and revised the description because it might not be critical in this context accordingly.

Fig5d: I think the anti-myc-Ub blot is missing to see where the 55 KD and 40 KD scale bands are.

Response: Thanks for this suggestion. We added 55 KD and 40 KD scale bands of the anti-myc-Ub blot in Fig. 5d.

Fig5e: It would have been interesting to add an RNase treatment to see if these interactions are RNA-dependent.

Response: Thanks for this suggestion. We performed co-IP assay by anti-METTL3 antibody with RNase A treatment and showed the interactions of METTL3 with DDB1 and SUV39H1/2 were decreased (Supplementary Fig. 8a).

REVIEWER COMMENTS

Reviewer #1 (Remarks to the Author):

Authors have addressed my comments.

Reviewer #2 (Remarks to the Author):

I thank the authors performed experiments to make this their conclusion solidier, but I still don't think my concerns are well addressed.

It seems there are no studied reports that METTL14 mutation (EPPL to APPA) could abolish METTL3-METTL14 m6A catalytic ability, so I still don't think this mutation is reasonable. It would be more reasonable to use METTL14 R298 and D312 mutations. Or at least it would be necessary for the authors to perform LC-MS to detect m6A abundance to make sure EPPL to APPA lost METTL3-METTL14 catalytic ability.

I still believe it would be necessary to knockout other writer proteins or eraser proteins to detect SUV39H1/2 and H3K9me3 levels.

If SUV39H1/2 regulation depends on m6A, but the reader protein YTHDC1 not required. Are other reader proteins required? In not, what's the role of m6A?

If METTL3/METTL14 promotes the degradation of H3K9me3 methyltransferase SUV39H1/2 only in H1 hESCs, not a general regulation. I believe it would be helpful for the authors put the data from HEK293T and HeLa cells in the manuscript and have a discussion.

I still confused why the down-regulated genes in METTL3 and METTL14 deletion hESCs were enriched in functions related to cellular metabolism, translation, rRNA processing as well as cell cycle progression. It would be helpful for the authors could perform SUV39H1/2 CHIP-seq to see if these regions have more SUV39H1/2 binding changed, or H3K9me3 CHIP-seq to see if these regions have more H3K9me3 changed.

It would be helpful for the authors to release the raw data deposited in original Figshare (<https://doi.org/10.6084/m9.figshare.25623738>).

Reviewer #3 (Remarks to the Author):

I would like to thank the authors for the substantial improvement of their manuscript. I feel that the concerns I raised have been satisfactorily addressed by additional experiments and

comments that strengthen the quality of the study's findings.
I believe it is now suitable for publication in Nature Communications

REVIEWER COMMENTS

Reviewer #1 (Remarks to the Author):

Authors have addressed my comments.

Response: We thank the reviewer for the positive comments.

Reviewer #2 (Remarks to the Author):

I thank the authors performed experiments to make this their conclusion solidier, but I still don't think my concerns are well addressed.

Response: We thank the reviewer for the positive comments.

It seems there are no studied reports that METTL14 mutation (EPPL to APPA) could abolish METTL3-METTL14 m⁶A catalytic ability, so I still don't think this mutation is reasonable. It would be more reasonable to use METTL14 R298 and D312 mutations. Or at least it would be necessary for the authors to perform LC-MS to detect m⁶A abundance to make sure EPPL to APPA lost METTL3-METTL14 catalytic ability.

Response: Thanks for this suggestion. We provided the immunostaining results for m⁶A abundance in rescued experiment using that expression of wild type while not EPPL-to-APPA mutant METTL14 rescue m⁶A abundance in *METTL14*-inducible knockout cells (*METTL14*-OE/KO hESCs) (see below, Supplementary Fig. 8f). These data clearly demonstrate that EPPL mutant METTL14 failed to induce m⁶A modification. As we responded in the 1st round revision, the major focus in current manuscript is to report a new role of METTL3/METTL14 as a scaffold for CRL4 complex rather than investigating the motifs for their m⁶A catalytic ability, which might be another research project worth to pursue in future.

I still believe it would be necessary to knockout other writer proteins or eraser proteins to detect SUV39H1/2 and H3K9me3 levels.

Response: Thanks for this suggestion. The major focus in current manuscript is on the new role of METTL3/METTL14 as a scaffold for CRL4 complex, rather than their m⁶A catalytic roles. In addition, the function of these other writer or eraser proteins for rRNA m⁶A has not been extensively examined so far, thus would be worth to pursue in future projects. We had a discussion in the revised manuscript.

If SUV39H1/2 regulation depends on m⁶A, but the reader protein YTHDC1 not required. Are other reader proteins required? In not, what's the role of m⁶A?

Response: Thanks for this suggestion. Based on current data, we are not sure how other reader proteins are involved in this process. RNA binding ability of METTL3/14 and RNA itself are essential for METTL3/14 to serve as the scaffold for CRL4 complex to facilitate SUV39H1/H2 degradation.

If METTL3/METTL14 promotes the degradation of H3K9me3 methyltransferase SUV39H1/2 only in H1 hESCs, not a general regulation. I believe it would be helpful for the authors put the data from HEK293T and HeLa cells in the manuscript and have a discussion.

Response: Thanks for this suggestion. Accordingly, we put the responding data from HEK293T and HeLa cells in the new Supplementary Fig. 8g-h and had a discussion in the revised manuscript.

I still confused why the down-regulated genes in METTL3 and METTL14 deletion hESCs were enriched in functions related to cellular metabolism, translation, rRNA processing as well as cell cycle progression. It would be helpful for the authors could perform SUV39H1/2 ChIP-seq to see if these regions have more SUV39H1/2 binding changed, or H3K9me3 CHIP-seq to see if these regions have more H3K9me3 changed.

Response: Thanks for this suggestion. Generally, nucleoli are fundamentally essential sites for ribosome biogenesis including rRNA synthesis/processing and ribosome assembly, and involve in diverse biological processes, such as cellular metabolism and cell cycle progression. While, nucleolar malfunction causes these processes to be disrupted and further induces various severe diseases. In current manuscript, these down-regulated genes enriched in cellular metabolism, translation, rRNA processing as well as cell cycle progression may be related to nucleolar malfunction induced by METTL3/METTL14 deficiency.

It would be helpful for the authors to release the raw data deposited in original Figshare (<https://doi.org/10.6084/m9.figshare.25623738>).

Response: Thanks for this suggestion. These data would be immediately released once the manuscript is accepted.

Reviewer #3 (Remarks to the Author):

I would like to thank the authors for the substantial improvement of their manuscript.

I feel that the concerns I raised have been satisfactorily addressed by additional experiments and comments that strengthen the quality of the study's findings.

I believe it is now suitable for publication in Nature Communications.

Response: We thank the reviewer for the positive comments.

REVIEWERS' COMMENTS

Reviewer #2 (Remarks to the Author):

I thank the authors for their best to address my concerns and the improvements of this manuscript. Generally, I believe this manuscript has been a strong candidate for Nature Communications. Congratulations.

REVIEWER COMMENTS

Reviewer #2 (Remarks to the Author):

I thank the authors for their best to address my concerns and the improvements of this manuscript. Generally, I believe this manuscript has been a strong candidate for Nature Communications. Congratulations.

Response: We thank the reviewer for the positive comments.